# Are nuclear masks all you need for improved out-of-domain generalisation? A closer look at cancer classification in histopathology

**Dhananjay Tomar**
University of Oslo
dhananjt@ifi.uio.no

**Alexander Binder**
Otto-von-Guericke University Magdeburg,
Singapore Institute of Technology
alexander.binder@ovgu.de

**Andreas Kleppe**[*]
Oslo University Hospital, University of Oslo,
UiT The Arctic University of Norway
andrekle@ifi.uio.no

## Abstract

Domain generalisation in computational histopathology is challenging because the images are substantially affected by differences among hospitals due to factors like fixation and staining of tissue and imaging equipment. We hypothesise that focusing on nuclei can improve the out-of-domain (OOD) generalisation in cancer detection. We propose a simple approach to improve OOD generalisation for cancer detection by focusing on nuclear morphology and organisation, as these are domain-invariant features critical in cancer detection. Our approach integrates original images with nuclear segmentation masks during training, encouraging the model to prioritise nuclei and their spatial arrangement. Going beyond mere data augmentation, we introduce a regularisation technique that aligns the representations of masks and original images. We show, using multiple datasets, that our method improves OOD generalisation and also leads to increased robustness to image corruptions and adversarial attacks. The source code is available at https://github.com/undercutspiky/SFL/

## 1 Introduction

Domain generalisation in histopathology is a crucial challenge because domain shifts naturally occur among hospitals and even within a single hospital or laboratory, e.g., temporally or among human operators and observers such as pathologists. Non-biological factors that substantially alter the images include differences in scanners, staining protocols, fixation of tissue, and even minor aspects like the manufacturer and storage conditions of stains [1].

Collecting data from numerous hospitals to address these domain shifts is often impractical and may not adequately reflect the full variability present in routine clinical practice, thus making it difficult to build computational histopathology models that generalise well. This leads us to focus on single-domain generalisation (S-DG) in this paper, specifically on how to train a model using data from only one hospital (considered a domain here) that generalises well to data from other hospitals. Popular S-GD methods in histopathology apply data augmentation and stain normalisation [2, 3]. The effectiveness of S-GD methods developed for natural images remains underexplored in histopathology [3]. Here, we compare these methods to a new, simple approach that we propose.

---

[*]Corresponding author.

38th Conference on Neural Information Processing Systems (NeurIPS 2024).

Research has shown that Convolutional Neural Networks (CNNs) tend to focus on texture over shape [4, 5]. However, in histopathology, the texture and colour of cell nuclei vary much more across domains than the shape and organisation of cell nuclei. As a result, focusing on shape features could improve a computational histopathology model's ability to generalise to unseen data because it may rely less on domain-specific features that vary across hospitals and more.

Nuclei in cancerous tissue exhibit distinct changes in shape, size, and overall organisation compared to nuclei in normal tissue [6–8]. Pathologists rely on these and other visual cues [9] for cancer diagnosis and grading, underscoring the biological importance and the consistency of nuclear morphology and organisation across domains. We hypothesise that focusing on nuclear morphology and organisation may be sufficient for cancer detection and that exploiting this during training could result in models with good generalisation.

We propose a method that encourages CNNs to focus more on nuclear morphology and organisation by using additional loss terms that prioritise shape-based features. Specifically, our method leverages nuclear segmentation masks during training to steer the learning towards nuclei. Through extensive experimentation, we demonstrate that this method improves performance on out-of-domain data without requiring nuclear segmentation masks at inference time, thus offering a promising and attractive solution for addressing domain generalisation in histopathology. Our contributions include:

- We propose a novel training method that incentivises the model to focus on nuclei.
- We evaluate our method on three datasets comprising hundreds of WSIs in total from various hospitals and organs. Our results show accuracy improvements over all other approaches.
- We evaluate the sensitivity of our method to image corruptions and adversarial attacks. Our results show performance improvements over the baseline.
- We conduct extensive ablation studies to show that models trained with our method focus on nuclei.

## 2   Related work

The prediction of various properties such as malignancy, grading, and HER2 expression using segmented nuclei has been a well-studied topic for many years [10–16]. Researchers have employed techniques such as watershed segmentation [17], thresholding, level sets [18], and snakes [19], often followed by extracting explicit morphometric features from the segmentations. For example, early work by Hasegawa et al. [20] focused on counting segmented regions, while Lee and Street [21] applied neural networks to the segmentation outputs. In contrast to these approaches, our method does not rely on segmentation during inference. Instead, we adjust the training process to encourage the extraction of nuclear features.

**Stain normalisation** methods convert the colours of a source image to match those of a target image. These methods were typically designed specifically for the most common type of histopathology images, which are images of tissue stained with haematoxylin and eosin (H&E). One of the earlier methods, Macenko normalisation [1], estimates stain vectors for source and target images and uses them to normalise the source image. Vahadane et al. [22] proposed a method that decouples stain "density maps" from "colour appearances", allowing the combination of the source image's density maps with the target image's colour appearances. Reinhard et al. [23] pioneered colour transfer by adjusting the global statistics of images in a different colour space, effectively transferring the colour characteristics of the target to the source image. Random Stain Normalization and Augmentation (RandStainNA) [24] combines stain normalisation and augmentation. Unlike traditional approaches that normalise using a fixed template, RandStainNA generates random virtual templates in the LAB [23] colour space and uses them to normalise the images during training. The templates are drawn from Gaussian distributions whose means and variances are derived from the training data. For a more comprehensive review, we refer the readers to [25]. In summary, the stain normalisation methods primarily focus on manipulating colour information to remove stain variability. On the other hand, our approach shifts the focus from colour manipulation to nuclear features.

**Data augmentation** is a common way to facilitate domain generalisation. Tellez et al. [2] evaluated several stain colour augmentation and stain normalisation methods and found that colour augmentation was crucial for good performance on external test sets in histopathology. Faryna et al. [26] extended RandAugment [27] by including certain histopathology-specific augmentations and excluding the ones that produce unrealistic-looking images. Tellez et al. [28] developed a data augmentation method

specific to H&E-stained images and used it for domain generalisation in mitosis detection. Pohjonen et al. [29] developed StrongAugment, where varying numbers of transformations are applied to an image to improve domain generalisation. Marini et al. [30] proposed Data-driven colour augmentation (DDCA), which evaluates an augmented image as acceptable or not for training based on its distance from other images in a database. Faryna et al. [31] evaluated different data augmentation strategies in histopathology, including manually selected augmentations, and found them all to be competitive.

**Single-domain generalisation (S-DG)** methods do not require data from multiple domains during training. Representation Self-Challenging (RSC) [32] works by discarding the features with relatively high gradients, making the model predict with the remaining features during training. Adversarial Domain Augmentation (ADA) [33] generates adversarial examples iteratively to augment the source domain and creates an ensemble of models. Meta-Learning-based ADA (M-ADA) [34] uses Wasserstein Auto-Encoder [35] to generate new samples and uses adversarial training on top along with a meta-learning scheme. Progressive domain expansion network (PDEN) [36] uses multiple autoencoders to generate new samples to expand the training set. Learning to Diversify (L2D) [37] introduces a learnable style-complement module that generates augmented images. The style-complement module is trained to diversify the images as much as possible but still keep the semantic information intact.

**Domain adaptation**, unlike S-DG, requires having access to some samples from the target domain. In histopathology, domain adaptation methods commonly make use of GAN [38] and CycleGAN [39]. StainGAN [40] uses CycleGAN to make images in the source domain look like the target domain. Residual CycleGAN [41] modifies the CycleGAN objective to have the generator produce the residual between domains instead of recreating the input image. In [42], authors augment a generator in CycleGAN with a stain colour matrix as an auxiliary input to stabilise the training. NST_AD_HRNet [43] uses Neural Style Transfer [44, 45] and GAN to preserve the content of the source image while combining it with the style of the target image. In some earlier works [46, 47], the input image is converted to greyscale and then coloured using a generator network which is based on a target image. While domain adaptation is not S-DG and thus a bit tangential to the focus of this paper, it is worth noting that domain adaptation is impractical in many clinical settings and may result in worse generalisation than stain normalisation and colour augmentation [48].

## 3  Proposed Method

Our approach aims to enhance S-DG by incentivising the model to focus on shaped-based features of nuclei in histopathological images and thereby reduce overfitting to irrelevant features that may carry higher label noise.

The first step involves generating segmentation masks that highlight specific areas of interest in the image. This step is applied only during training, while test-time evaluation relies solely on H&E-stained images. As we hypothesised that nuclear morphology and organisation contain sufficient information for cancer detection, our segmentation masks are binary images with nuclear pixels as foreground and other pixels as background.

One possible approach to using the segmentation mask is to include it as a fourth channel in the input image. Alternatively, the mask can be used as the sole input to the model. However, both methods necessitate running the segmentation model during inference, which increases computational demands and slows down processing.

Our method circumvents the need for a nuclear segmentation network at inference time by incorporating additional loss terms during training. For a given input image $x$ and its corresponding segmentation mask $x'$, our method involves the following steps:

1. Execute a forward propagation through the neural network model on both the H&E-stained image $x$ and its nuclear mask $x'$, saving the embeddings generated by the network as $z$ and $z'$ for $x$ and $x'$, respectively.

2. Compute the Binary Cross-Entropy (BCE) loss for both $x$ and $x'$.

3. Compute the $\ell_2$-distance between the embeddings $z$ and $z'$.

4. Minimise the sum of the two CE losses and the $\ell_2$-distance.

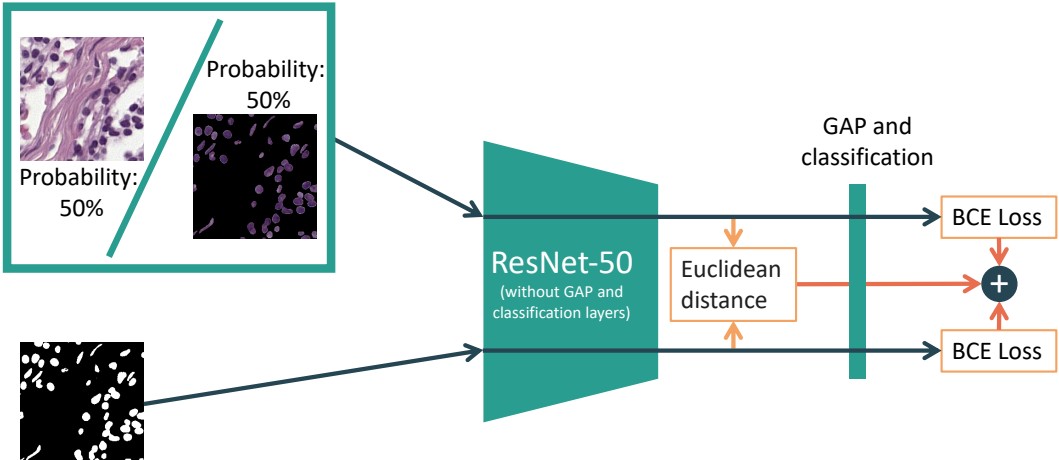

Figure 1: We pass the input image (or, with 0.5 probability, input image multiplied with its nuclear segmentation mask) and its nuclear segmentation mask through the network and minimise the Binary Cross-Entropy (BCE) loss for both the input image and its mask. Additionally, we minimise the $\ell_2$-distance between the input image's embedding vector and the mask's embedding vector just before the Global Average Pooling (GAP) layer. The embedding vector is ResNet-50's penultimate layer's feature map, i.e., stage 4's last feature map.

Our approach is illustrated in Figure 1. We employ a ResNet-50 [49] from Torchvision [50] as the base model. We next discuss some details of our approach.

$\ell_2$-**regularisation:** To encourage the network to focus on nuclei, we minimise the distance between the feature map of the original image and that of its nuclear segmentation mask. We use the flattened feature map from ResNet-50's penultimate layer, just before Global Average Pooling, to obtain the embeddings. The regularisation term consists of the $\ell_2$-distance between the embeddings of the original image $z$ and its mask $z'$, which is added to the BCE losses for both the image and the mask. Let $\hat{y}$ be the model's prediction for $x$, $\hat{y}'$ for $x'$, and $y$ be the ground truth. Then, the total loss $L$ is:

$$L = \lambda \|z - z'\|_2^2 + BCE(y, \hat{y}) + BCE(y, \hat{y}') \tag{1}$$

where $BCE$ is the Binary Cross-Entropy loss function for labels in $\{0, 1\}$:

$$BCE(y, p) = -(y \log(p) + (1 - y) \log(1 - p)) \tag{2}$$

**Original image times mask:** Since the embeddings of the original image and its binary segmentation mask may differ significantly, minimizing their $\ell_2$-distance can be challenging for the model. To address this, with a probability of 0.5, we multiply the original image by its segmentation mask, i.e., the network receives $x * x'$ as input half the time instead of $x$. By multiplying with the segmentation mask, everything in the original image except the nuclei is set to 0. Figure 1 shows what the output looks like. By simplifying the task, the network can more easily reduce the distance between the embeddings of the nuclei-only image and the mask and gradually improve alignment between the embeddings of the original image and the mask. We found this augmentation to help stabilise training.

## 4 Experiments

### 4.1 Datasets

**CAMELYON17** [51] dataset consists of 1000 H&E-stained Whole Slide Images (WSIs) of breast cancer metastases in lymph node sections from five medical centres in the Netherlands. It contains pixel-level annotations of tumours for 10 WSIs from each medical centre, giving us 50 WSIs to work with. WSIs from centres 0, 3 and 4 were scanned using the same scanner, while the other two centres used a different scanner each. All slides were scanned at $40\times$ resolution. We treat each centre as a different domain.

**BCSS** [52] dataset consists of 151 H&E-stained WSIs of histologically-confirmed primary breast cancer cases from The Cancer Genome Atlas (TCGA) with triple-negative status determined from

clinical data files. All WSIs have a resolution of $40\times$. The WSIs were annotated at the pixel level using crowdsourcing. Each pixel can have one of the many labels. We consider the label "tumor" to define pixels with a tumour and all other labels except "outside_roi", "exclude", or "undetermined" to define pixels without a tumour.

**Ocelot** [53] dataset consists of pixel-level annotations of tumour vs non-tumour pixels for 303 WSIs from TCGA. It consists of WSIs of primary tumour from six different organs: Bladder, Endometrium, Head-and-neck, Kidney, Prostate, and Stomach. The annotations in the dataset are at a low resolution, so we upscale the annotations to $40\times$. We exclude two WSIs that have only $20\times$ resolution; all other WSIs have $40\times$ resolution.

## 4.2 Dataset preparation

We use code from WILDS [54, 55] to prepare the CAMELYON17 dataset with modifications. Tiles are sized $(270 \times 270)$ at $40\times$ resolution. For each domain (medical centre), data is split by patient, ensuring all tiles from a patient are in a single subset. Since the number of tiles varies drastically across patients, we shuffle patients so that the validation subset contains 20%-25% of all tiles per domain. Our processed version of CAMELYON17 is available at [56].

## 4.3 Experiment setup

We train models using the CAMELYON17 dataset, treating each medical centre as a distinct domain. We use the BCSS and Ocelot datasets as external test datasets. To avoid multiple comparisons and overly optimistic performance estimates, we use the external test datasets only once during the entire project, solely to evaluate the final models [57].

For each combination of medical centre and method, we train ten models using the train subset of that centre. Thus, we train 50 models in total for each method. We use the loss on the validation dataset (of the training domain) to select the best model for each training. All models are trained for 50 epochs using the Adam optimiser with a learning rate $4\mathrm{e}{-}5$ and a weight decay of $1\mathrm{e}{-}4$. We use exponential learning rate decay with a decay rate of $0.955$. For our method, we set the parameter $\lambda$ in equation (1) to $\lambda = 0$ for the first five epochs, effectively training without $\ell_2$-distance loss in these epochs, and then use $\lambda = 1$ for the rest of the training. We start saving models for selection of the one with lowest validation loss after ten epochs for our method to allow the network to stabilise while from the first epoch for other methods. For all experiments, unless stated otherwise, we use a ResNet-50 [49] model pre-trained on ImageNet [58]. We use HoVer-Net [59] trained on the CoNSeP [59] dataset to generate nuclear segmentation masks.

While domain generalisation encompasses a wide variety of methods, we have selected several exemplary baselines for comparison: Macenko normalisation [1], RSC [32], L2D [37], RandStainNA (RandSNA in result tables) [24], and DDCA [30]. We also include a baseline where we initialise ResNet-50 with pre-trained weights from HoVer-Net [59]. These methods represent different approaches, including stain normalisation (Macenko, RandStainNA) and generating augmented images (L2D). By selecting these diverse techniques, we ensure a comprehensive evaluation of our method's performance across various S-DG strategies.

It is important to note that our method can be integrated with many existing S-DG approaches, making it a flexible plug-in solution rather than a direct competitor. We evaluate most methods with and without the photometric augmentations selected for ERM. After testing various augmentation strategies available in Torchvision [50], we identified the most effective combination to be: ColorJitter(brightness=[0.5, 1.5], contrast=[0.5, 1.5], saturation=[0.5, 1.5], hue=[-0.3, 0.3]) and GaussianBlur(kernel_size=3). Results using these augmentations are marked as '-Aug' in the results tables. In all experiments, including those without photometric augmentations, we apply the basic geometric augmentations: random horizontal and vertical flips. For all ViT-Tiny [60] experiments, we also add affine augmentations: random rotation (up to 90°) and translation (up to 45 pixels).

We ran the experiments on two clusters with GPUs with 64 GB (AMD MI250X) and 24 GB (Nvidia RTX 3090) GPU RAM each. Each job consumed about 21 to 31 GB of GPU RAM. The proposed method took 5 to 20 hours to train, depending on the train data size while ERM took 2.5 to 11 hours.

We report tile-level accuracy for tumour vs non-tumour tile classification for all datasets. Additionally, we measure robustness to image noise by measuring the accuracy drop on CAMELYON17 for image

Table 1: Out-of-domain accuracy on CAMELYON17. The column name indicates the centre used to train models. The best accuracy for each column is in **bold face** and the second best in *italics*. Method "Ours-no-$\ell_2$-A" is shorthand for "Ours-no-$\ell_2$-Aug" and refers to our approach without $\ell_2$-regularisation. Method "Ours-MO-Aug" refers to our approach with masks only, that is, neither using $\ell_2$-regularisation nor using mask-times-input augmentation of H&E images with 50% probability during training. A paired t-test for "L2D-Aug" versus "Ours-Aug" yields a p-value of $2 \cdot 10^{-5}$.

| Method | Centre-0 | Centre-1 | Centre-2 | Centre-3 | Centre-4 | Average |
|---|---|---|---|---|---|---|
| ERM | $72.8 \pm 2.3$ | $65.9 \pm 3.7$ | $64.1 \pm 3.0$ | $55.0 \pm 1.3$ | $53.8 \pm 3.0$ | $62.4 \pm 2.7$ |
| Macenko | $79.3 \pm 2.1$ | $62.4 \pm 1.4$ | $73.3 \pm 5.0$ | $65.8 \pm 2.3$ | $85.9 \pm 4.2$ | $73.3 \pm 3.0$ |
| HoVerNet | $72.5 \pm 2.4$ | $71.0 \pm 2.5$ | $61.3 \pm 3.9$ | $55.1 \pm 1.9$ | $49.6 \pm 8.4$ | $61.9 \pm 3.8$ |
| RandSNA | $75.7 \pm 3.1$ | $70.9 \pm 4.9$ | $62.4 \pm 2.5$ | $57.2 \pm 2.8$ | $51.8 \pm 2.6$ | $63.6 \pm 3.2$ |
| RSC | $77.1 \pm 3.2$ | $64.5 \pm 3.1$ | $61.9 \pm 3.8$ | $56.8 \pm 2.3$ | $51.1 \pm 2.2$ | $62.3 \pm 2.9$ |
| L2D | *$93.6 \pm 1.0$* | $72.9 \pm 2.5$ | $64.4 \pm 13.0$ | $73.6 \pm 4.3$ | $84.4 \pm 3.7$ | $77.8 \pm 4.9$ |
| Ours | $90.4 \pm 1.5$ | **$92.5 \pm 0.3$** | $90.1 \pm 1.3$ | $82.1 \pm 2.7$ | $90.8 \pm 1.0$ | *$89.2 \pm 1.3$* |
| ERM-Aug | $93.1 \pm 1.0$ | $78.9 \pm 2.1$ | $89.3 \pm 2.8$ | $74.8 \pm 1.5$ | $91.3 \pm 1.6$ | $85.5 \pm 1.8$ |
| Macenko-Aug | $86.3 \pm 1.9$ | $78.7 \pm 1.5$ | $86.2 \pm 4.4$ | $70.0 \pm 2.8$ | $90.8 \pm 1.2$ | $82.4 \pm 2.3$ |
| HoVerNet-Aug | $93.0 \pm 0.6$ | $80.8 \pm 2.8$ | *$91.3 \pm 1.2$* | $82.2 \pm 1.6$ | $89.6 \pm 2.2$ | $87.4 \pm 1.7$ |
| RandSNA-Aug | $92.7 \pm 1.1$ | $83.1 \pm 2.1$ | $91.0 \pm 2.0$ | $78.9 \pm 3.0$ | $91.1 \pm 1.5$ | $87.4 \pm 1.9$ |
| DDCA-Aug | $92.5 \pm 2.4$ | $79.4 \pm 1.9$ | $89.4 \pm 2.9$ | $78.2 \pm 3.1$ | $90.2 \pm 2.1$ | $86.0 \pm 2.5$ |
| RSC-Aug | $93.1 \pm 0.8$ | $78.2 \pm 2.0$ | $89.3 \pm 3.4$ | $77.9 \pm 2.2$ | $91.0 \pm 1.7$ | $85.9 \pm 2.0$ |
| L2D-Aug | **$94.3 \pm 0.1$** | $87.6 \pm 0.6$ | $87.7 \pm 1.4$ | *$83.4 \pm 2.6$* | **$92.3 \pm 0.9$** | $89.1 \pm 1.1$ |
| Ours-Aug | $91.8 \pm 0.7$ | *$92.2 \pm 1.6$* | **$92.9 \pm 0.7$** | **$90.4 \pm 1.1$** | *$91.7 \pm 0.5$* | **$91.8 \pm 0.9$** |
| Ours-no-$\ell_2$-A | $92.1 \pm 0.8$ | $81.4 \pm 2.9$ | $91.8 \pm 2.0$ | $83.3 \pm 1.0$ | $90.5 \pm 1.7$ | $87.8 \pm 1.7$ |
| Ours-MO-Aug | $91.8 \pm 2.2$ | $88.2 \pm 2.1$ | $85.0 \pm 2.2$ | $79.7 \pm 4.2$ | $77.9 \pm 2.8$ | $84.5 \pm 2.7$ |

corruptions introduced in [61]. This includes Gaussian-, shot-, impulse- and snow-noise, and two blur types, elastic transform and JPEG compression.

**Results on CAMELYON17 (lymph node sections)**    We test models on their respective out-of-domain data. E.g., a model trained on Centre-3 is tested on all the data from Centre-0,1,2,4. Our method attains 10% higher accuracy than the next best method (L2D) when none used photometric augmentations and was also superior when photometric augmentations were used (Table 1).

**Results on BCSS (primary breast cancer)**    The accuracy of the models trained on a centre in CAMELYON17 drops substantially (12% to 14%) for all the methods when tested on BCSS (Table 2) compared to when tested on other centres in CAMELYON17 (Table 1). This could be due to a mismatch between pathologists' annotations on CAMELYON17 and BCSS but also due to the biological differences between these tissue types. In particular, epithelial cells in lymph nodes would almost certainly be tumour cells, while they could be benign cells in ordinary breast tissue. These results show that the relative performance on CAMELYON17 for different methods is indicative of relative performance on an external test set, as the performance drop is similar for all methods.

**Results on Ocelot (primary non-breast cancer)**    We test our model on the Ocelot dataset to evaluate if our method helps to train models that generalise to other organs as well. Ocelot does not have any data from breast tissue nor does it include lymph node sections (which all the models have been trained on). We report the results of these experiments in Table 3. While our method achieves the highest accuracy also in this case, the difference between our method and L2D is not as big as it is for CAMELYON17 and BCSS. Taking a closer look into the performance for separate organs (Tables 4, 5, 6, 7, 8, and 9 in the Supplement), we can see that our method performs worse than L2D in Endometrium and Kidney, where accuracies are generally lower, and better in the four other organs. This indicates that models trained with our method generalise worse to organs where the transferability from breast tissue is generally low. This is at least the case for Kidney which has by far the lowest accuracies across all methods. Generalising to different cancer types is an emerging experimental topic; see, for example, [62].

In summary, our method yields better accuracy than the baselines, including other S-DG approaches.

Table 2: Out-of-domain accuracy on BCSS. The column name indicates the centre used to train models. The best accuracy for each column is in **bold face** and the second best in *italics*. A paired t-test for "L2D-Aug" versus "Ours-Aug" yields a p-value of $4 \cdot 10^{-5}$.

| Method | Centre-0 | Centre-1 | Centre-2 | Centre-3 | Centre-4 | Average |
|---|---|---|---|---|---|---|
| ERM | $58.0 \pm 2.9$ | $69.5 \pm 2.8$ | $52.0 \pm 1.0$ | $50.0 \pm 0.1$ | $52.1 \pm 3.9$ | $56.3 \pm 2.1$ |
| Macenko | $68.7 \pm 2.4$ | $56.5 \pm 1.9$ | $65.5 \pm 2.6$ | $56.8 \pm 2.3$ | $71.1 \pm 3.4$ | $63.7 \pm 2.5$ |
| HoVerNet | $55.7 \pm 2.0$ | $65.3 \pm 2.2$ | $53.8 \pm 0.8$ | $49.8 \pm 1.9$ | $47.5 \pm 2.9$ | $54.4 \pm 2.0$ |
| RandSNA | $60.5 \pm 2.8$ | $67.4 \pm 3.8$ | $51.0 \pm 0.8$ | $50.1 \pm 0.7$ | $50.2 \pm 1.6$ | $55.8 \pm 1.9$ |
| RSC | $63.3 \pm 3.7$ | $65.7 \pm 2.4$ | $50.8 \pm 1.0$ | $50.1 \pm 0.1$ | $50.2 \pm 0.5$ | $56.0 \pm 1.5$ |
| L2D | $79.9 \pm 1.2$ | $65.2 \pm 0.7$ | $67.4 \pm 2.3$ | $63.2 \pm 4.5$ | $66.2 \pm 2.2$ | $68.4 \pm 2.2$ |
| Ours | $74.2 \pm 3.6$ | *$78.3 \pm 2.2$* | $73.4 \pm 1.9$ | $63.8 \pm 2.8$ | $71.8 \pm 2.3$ | $72.3 \pm 2.6$ |
| ERM-Aug | $80.1 \pm 1.6$ | $70.7 \pm 2.8$ | $73.7 \pm 2.6$ | $60.9 \pm 1.8$ | $73.3 \pm 2.6$ | $71.7 \pm 2.3$ |
| Macenko-Aug | $75.8 \pm 2.9$ | $67.6 \pm 2.4$ | $72.6 \pm 2.6$ | $57.8 \pm 2.9$ | *$75.3 \pm 1.8$* | $69.8 \pm 2.5$ |
| HoVerNet-Aug | $79.8 \pm 1.4$ | $65.1 \pm 1.6$ | $71.2 \pm 2.2$ | $64.0 \pm 2.2$ | $69.2 \pm 6.1$ | $69.9 \pm 2.7$ |
| RandSNA-Aug | $78.5 \pm 2.7$ | $73.2 \pm 3.1$ | $72.8 \pm 3.2$ | $64.5 \pm 3.4$ | $75.1 \pm 2.5$ | $72.8 \pm 3.0$ |
| DDCA-Aug | $79.1 \pm 1.9$ | $71.7 \pm 3.5$ | $70.1 \pm 2.8$ | $61.4 \pm 3.2$ | $71.4 \pm 7.3$ | $70.7 \pm 3.8$ |
| RSC-Aug | $79.6 \pm 1.5$ | $72.1 \pm 2.9$ | $71.6 \pm 1.7$ | $63.2 \pm 1.6$ | $74.1 \pm 4.3$ | $72.1 \pm 2.4$ |
| L2D-Aug | *$81.9 \pm 0.3$* | $74.7 \pm 0.6$ | *$74.2 \pm 0.6$* | *$67.5 \pm 2.9$* | **$77.2 \pm 2.2$** | *$75.1 \pm 1.3$* |
| Ours-Aug | **$82.3 \pm 0.8$** | **$81.9 \pm 2.3$** | **$75.7 \pm 2.2$** | **$79.6 \pm 1.0$** | $74.8 \pm 1.9$ | **$78.8 \pm 1.6$** |

Table 3: Out-of-domain accuracy on Ocelot. The column name indicates the centre used to train models. The best accuracy for each column is in **bold face** and the second best in *italics*. A paired t-test for "L2D-Aug" versus "Ours-Aug" yields a p-value of $0.044$.

| Method | Centre-0 | Centre-1 | Centre-2 | Centre-3 | Centre-4 | Average |
|---|---|---|---|---|---|---|
| ERM | $65.7 \pm 2.4$ | $55.8 \pm 1.7$ | $51.7 \pm 1.3$ | $45.6 \pm 1.4$ | $53.5 \pm 5.0$ | $54.5 \pm 2.4$ |
| Macenko | $64.3 \pm 1.9$ | $54.4 \pm 0.9$ | $63.8 \pm 1.4$ | $54.8 \pm 2.0$ | $65.3 \pm 3.7$ | $60.5 \pm 2.0$ |
| HoVerNet | $62.0 \pm 1.2$ | $53.7 \pm 2.2$ | $52.1 \pm 0.7$ | $48.0 \pm 1.9$ | $54.3 \pm 2.9$ | $54.0 \pm 1.8$ |
| RandSNA | $67.0 \pm 2.5$ | $56.1 \pm 1.9$ | $51.0 \pm 0.3$ | $46.3 \pm 1.9$ | $51.2 \pm 2.4$ | $54.3 \pm 1.8$ |
| RSC | $68.1 \pm 2.4$ | $55.6 \pm 1.0$ | $50.9 \pm 0.6$ | $47.2 \pm 1.5$ | $50.3 \pm 0.8$ | $54.4 \pm 1.3$ |
| L2D | $68.2 \pm 1.3$ | $57.3 \pm 0.5$ | $56.4 \pm 2.4$ | $55.9 \pm 3.3$ | $60.1 \pm 4.0$ | $59.6 \pm 2.3$ |
| Ours | $67.9 \pm 1.4$ | *$70.7 \pm 1.0$* | $66.7 \pm 1.2$ | $62.1 \pm 1.8$ | $69.2 \pm 0.9$ | $67.3 \pm 1.3$ |
| ERM-Aug | *$74.0 \pm 1.4$* | $62.8 \pm 1.9$ | $67.6 \pm 2.5$ | $56.0 \pm 1.6$ | $67.6 \pm 3.1$ | $65.6 \pm 2.1$ |
| Macenko-Aug | $68.8 \pm 2.0$ | $60.9 \pm 1.1$ | **$70.3 \pm 1.5$** | $57.6 \pm 3.2$ | *$72.5 \pm 1.7$* | $66.0 \pm 1.9$ |
| HoVerNet-Aug | $70.7 \pm 1.0$ | $54.2 \pm 1.9$ | *$69.4 \pm 2.3$* | $61.2 \pm 2.4$ | $69.2 \pm 2.9$ | $65.0 \pm 2.1$ |
| RandSNA-Aug | $70.8 \pm 3.2$ | $66.8 \pm 2.5$ | $68.4 \pm 2.5$ | $61.3 \pm 2.5$ | $71.7 \pm 2.7$ | $67.8 \pm 2.7$ |
| DDCA-Aug | $73.1 \pm 2.4$ | $64.9 \pm 2.5$ | $68.9 \pm 2.7$ | $57.4 \pm 2.9$ | $66.9 \pm 4.5$ | $66.2 \pm 3.0$ |
| RSC-Aug | $71.9 \pm 2.2$ | $61.7 \pm 2.4$ | $69.2 \pm 2.9$ | $58.4 \pm 1.4$ | $70.1 \pm 3.6$ | $66.3 \pm 2.5$ |
| L2D-Aug | **$74.7 \pm 0.6$** | $68.3 \pm 0.5$ | $65.6 \pm 0.7$ | *$62.5 \pm 2.7$* | **$74.4 \pm 1.2$** | *$69.1 \pm 1.1$* |
| Ours-Aug | $70.8 \pm 0.5$ | **$72.0 \pm 1.3$** | $68.9 \pm 1.3$ | **$70.4 \pm 0.7$** | $70.7 \pm 1.3$ | **$70.6 \pm 1.0$** |

## 5   Ablation Study and Discussion

**Impact of data augmentation**   Tables 1, 2, and 3 shows that data augmentation benefits all methods substantially, which is consistent with well-established knowledge.

**Impact of $\ell_2$-regularisation**   The result labelled *Ours-no-$\ell_2$-A* in Table 1 shows that using data augmentation with nuclear masks alone is insufficient to achieve high accuracy. Without $\ell_2$-regularisation (i.e., setting $\lambda = 0$ in Equation (1)), our method only slightly outperforms most baselines that also uses data augmentation. The key factor for effective cross-domain generalisation is the ability to align the feature representation of input images with corresponding mask images, which lack colour and texture. Further evidence supporting this alignment effect is presented in the next paragraph.

**Impact of mask-times-input-augmentation**   The result *Ours-MO-Aug* in Table 1 demonstrates an ablation with two changes: the absence of $\ell_2$-regulation and the removal of the 50% probability

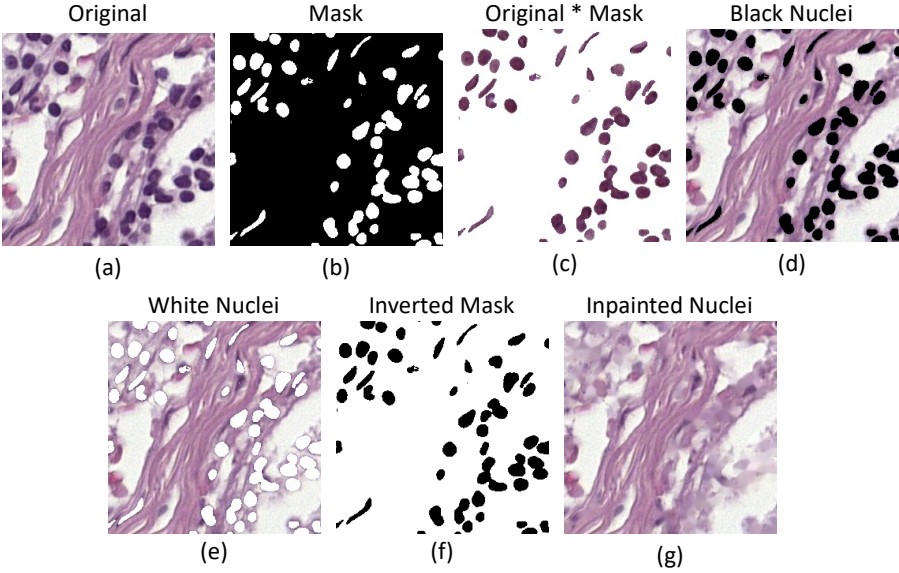

Figure 2: Exemplary image ablations used in this study.

mask-times-input augmentation of H&E images during training (Figure 1), but still having two CE loss terms, one of them over nuclear masks. We see a further decline of accuracies below most baselines with photometric augmentations.

**Impact of learned features without nuclear-mask-like features**   Here, we bring further evidence for the effect of $\ell_2$-regularisation about pulling the features towards the representation of nuclear masks. Note that nuclear masks are not used during inference in standard evaluations such as all those in Tables 1, 2, and 3. In Table 10 in the Supplement, we can see results for predictions in which the embeddings (features before the GAP layer) of the H&E images are modified by subtracting the embeddings of the corresponding nuclear masks. By comparing to Table 1, we see a drop in performance for all methods. However, the drop is largest for our method, with an accuracy below random guessing. This shows that the features computed from H&E-stained images at test time are indeed more similar to features from nuclear masks for our method than for other methods.

**Impact of removal of intranuclear texture and colour**   In Tables 11 and 14 in the Supplement, we consider the performance on modified H&E images, in which intranuclear texture is removed by masking it out with a constant colour (see examples in Figure 2d and 2e). This is of interest due to the observation that intranuclear texture is often different in cancerous nuclei, which can be informative to humans. We can see that if it is replaced by a colour similar to the colour of nuclei, we for our method obtain a performance (Table 11) very similar to the performance with original H&E images (Table 1). On the other hand, changing the colour to white seems to reduce the performance notably (Table 14). This is possibly due to the creation of images with outlier statistics. A more likely explanation is that it is common for H&E stains to have small holes or gaps of white background colour in the stroma, which usually are not discriminative information but rather shear stress artefacts from the tissue cutting process. Therefore, masking nuclei with white masks may effectively remove discriminative information about nuclei. This domain-specific observation may explain the asymmetry in behaviour when masking nuclei with black versus white.

**Impact of removal of extranuclear information**   Table 23 in the Supplement shows results on data where all the non-nuclear background is set to white (see example in Figure 2c). These images can be viewed as an inside-out inverted case of the images evaluated to give the results in Table 11. The common information in both sets of images is the morphology and organisation of nuclei. The performance for our proposed method remains high on these images (Table 23), being close to the best result on original H&E data (Table 1). The experiments in Tables 23 and 11 demonstrate the strong generalisability of focusing on nuclear morphology and organisation in out-of-domain settings.

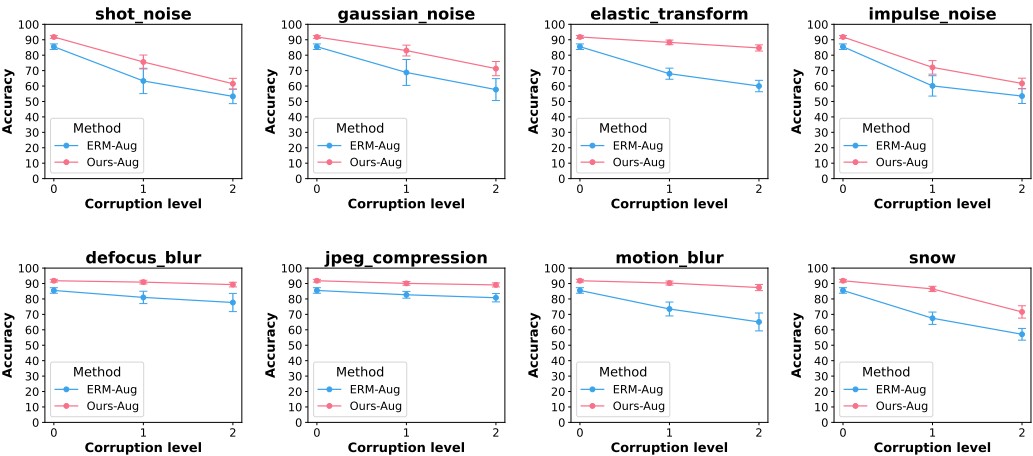

Figure 3: Robustness to added noise described in [61].

**Impact of dilution of nuclear shapes**   We expand nuclei masks by a classic morphological dilation and then blacken the dilated regions in the H&E images. The nuclear shape information in these images is thus progressively reduced compared to the images where only the nuclei are filled with black. Across all methods, we observe a drop in accuracy with an increase in dilation (Tables 11, 12, and 13 in the Supplement), highlighting the critical role of shape in this domain. The proposed method is more robust to moderate shape dilution with a mask size of 5 than the baseline methods. A similar but stronger trend appears in Tables 14, 15, and 16 in the Supplement for whitened nuclei.

**Impact of removing nuclei**   We dilate the nuclear mask image with a kernel size of 5 to encapsulate remnants of the boundary of nuclei and then use the dilated mask to remove nuclei by inpainting [63]. The accuracy with the resulting images (see example in Figure 2g) drops to random guessing for our method (Tables 17 and 18 in the Supplement). Essentially no tiles are classified as tumour, giving a nearly zero recall and low precision (Tables 19, 20, 21, and 22 in the Supplement). Also, this supports that models trained using our method focus on nuclear morphology and organisation, and shows that the models reasonably associate the absence of nuclei with no tumour.

**Saliency maps via Integrated Gradients**   To further demonstrate that our method steers models to focus on nuclei, we generate saliency (pixel attribution) maps using Integrated Gradients [64] and show some randomly selected examples in Figures 6,7 in the Supplement. The saliency maps also indicate that a model trained using our method focuses on nuclei.

**Evaluation of L2D and RSC combined with the proposed method**   Tables 26, 27, and 28 in the Supplement show the results of combining L2D and RSC with the proposed method. Combining the proposed method, which regularises, with L2D, which diversifies, yields mixed results, likely due to the opposing effects of these two interventions. Combining it with RSC results in a small gain over using our method alone. Overall, this demonstrates the effectiveness of the method proposed.

**Evaluation on segmentation mask data**   For the sake of completeness, we show in Tables 24 and 25 in the Supplement that our method also performs well when tested on nuclear masks (as exemplified in Figure 2b) and their inversions (see example in Figure 2f).

**Evaluation of robustness to image corruptions**   Figure 3 shows that the proposed method has notably higher robustness to image corruptions for most experiments in eight types of corruptions described in [61]. Samples of corrupted images are shown in Figure 5 in the Supplement.

**Evaluation of robustness against adversarial attacks**   We evaluated the robustness of models against adversarial attacks [65] using the Projected Gradient Descent (PGD) attack [66]. Figure 4a demonstrates that models trained using our method have significantly higher robustness than ERM and L2D, the latter being the second-best performing method in Tables 1, 2, and 3. Additionally, we

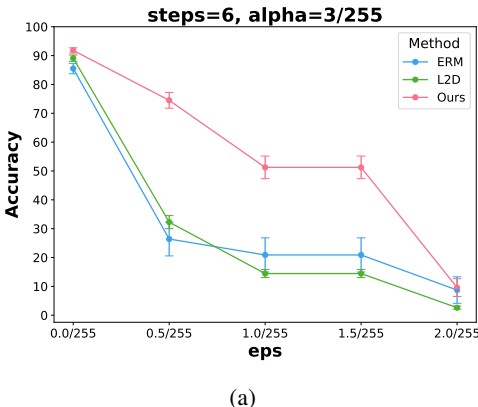
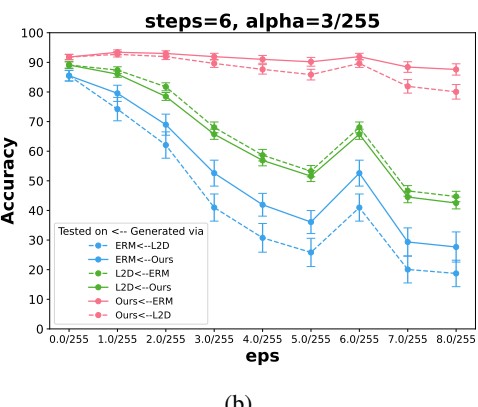

(a)             (b)

Figure 4: **(a)** PGD attack on models. **(b)** Cross-model PGD attacks where adversarial images are generated using a model from a method but the accuracy for those images is tested on models from other methods. Results are for the validation subset of each centre in CAMELYON17.

conduct cross-model attacks by generating adversarial images using models trained with one method and evaluating them on models trained with other methods. The results indicate that our models exhibit minimal performance degradation when exposed to adversarial images generated by models from other methods (Figure 4b). In contrast, the accuracy of models trained with ERM and L2D drops substantially. This further demonstrates the superior robustness of models from our method.

**A preliminary evaluation on a transformer architecture** We perform a comparison using a fine-tuned ViT-Tiny [60] model. The results are shown in Tables 29, 30 and 31 in the Supplement. These results show that our approach obtains superior out-of-domain performance for the CAMELYON17 dataset. The results are more mixed for the other datasets. In particular, it seems that models trained on one of the five centres (Center-4) in CAMELYON17 do not generalise well to other cancer types and are actually also performing sub-optimally in CAMELYON17. For models trained on each of the other four centres in CAMELYON17, the performance with our approach is, on average, better than with other approaches, but the performance increase is lower than for ResNet-50. However, in the same tissue type (CAMELYON17 data), the performance gain is similar for both ViT-Tiny and ResNet-50. Our interpretation of all these results is that our approach can improve out-of-domain performance also for ViT-Tiny, in particular across centres and scanners for the same tissue type, but that it might also fail for a minority of the training datasets. This experiment is preliminary because we took the same hyperparameters as used for ResNet-50, including the same learning rate and $\lambda = 1$, both of which might not be optimal. Also, we note that ViT-Tiny has much fewer parameters than ResNet-50. Experiments with larger transformers might obtain bigger differences, as seen in [67].

**Limitations of this study** As a limitation, we identify that we have performed these experiments for only one classification task. For medical practitioners, it would be of interest to measure the impact for other tasks, such as tumour grading and survival prediction, when evaluated in an out-of-domain generalisation setup. However, this would require access to multi-centre datasets with relevant labels available. Secondly, we ran the full set of experiments only on one base network, ResNet-50, because we preferred to run a larger set of ablation experiments to understand what actually has been learned when using our method. While we expect results to be qualitatively similar for other CNNs, transformer networks might have different learning dynamics, and results for those with a larger capacity than the ViT-Tiny are of interest in future work. Finally, an extension to other cancer types, such as prostate or colon cancer, would also be of interest.

## 6 Conclusion

We have shown a simple method to enforce the learning of shape features at training time, which uses unmodified input images at inference time. It shows very good out-of-domain performance and can be combined as a plugin with other methods to enhance out-of-domain generalisation. Aside from out-of-domain accuracy, the proposed method gives improved robustness to image alterations.

## Acknowledgments and Disclosure of Funding

The computations were performed on resources provided by Sigma2—the National Infrastructure for High-Performance Computing and Data Storage in Norway—through Project NN8104K. Additionally, we acknowledge Sigma2 for access to the LUMI supercomputer, owned by the EuroHPC Joint Undertaking, hosted by CSC (Finland) and the LUMI consortium through Sigma2, Norway, Project 465000262. This work was supported by the authors' institutions, the Research Council of Norway (project number 309439), the National Research Foundation, Singapore, and Infocomm Media Development Authority under its Trust Tech Funding Initiative. Any opinions, findings, and conclusions or recommendations expressed in this work are those of the authors and do not reflect the views of their institutions, the Research Council of Norway, the National Research Foundation, Singapore, and Infocomm Media Development Authority.

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

# Technical Appendix / Supplement

Table 4: Out-of-domain accuracy on Ocelot evaluated only on the organ BLADDER. The column name indicates the centre used to train models. The best accuracy for each column is in **bold face** and the second best in *italics*.

| Method | Centre-0 | Centre-1 | Centre-2 | Centre-3 | Centre-4 | Average |
|--------|----------|----------|----------|----------|----------|---------|
| ERM | $65.0 \pm 3.1$ | $58.2 \pm 3.5$ | $50.4 \pm 0.6$ | $46.4 \pm 1.8$ | $53.1 \pm 6.9$ | $54.6 \pm 3.2$ |
| Macenko | $71.8 \pm 1.4$ | $58.7 \pm 2.0$ | $66.2 \pm 2.1$ | $60.4 \pm 3.0$ | $65.6 \pm 5.8$ | $64.6 \pm 2.9$ |
| RSC | $70.3 \pm 3.0$ | $58.0 \pm 2.7$ | $50.2 \pm 0.1$ | $47.8 \pm 1.4$ | $50.0 \pm 0.9$ | $55.3 \pm 1.6$ |
| L2D | $72.0 \pm 1.5$ | $59.5 \pm 1.4$ | $59.5 \pm 2.2$ | $58.2 \pm 4.8$ | $58.4 \pm 4.1$ | $61.5 \pm 2.8$ |
| Ours | $73.9 \pm 2.7$ | *$74.9 \pm 1.7$* | *$74.6 \pm 1.8$* | *$64.3 \pm 3.3$* | *$74.0 \pm 1.0$* | *$72.4 \pm 2.1$* |
| ERM-Aug | *$76.6 \pm 1.9$* | $68.3 \pm 2.8$ | $63.9 \pm 3.3$ | $57.0 \pm 1.9$ | $63.7 \pm 5.0$ | $65.9 \pm 3.0$ |
| Macenko-Aug | $71.9 \pm 2.3$ | $65.9 \pm 1.1$ | $70.3 \pm 2.4$ | $60.0 \pm 3.6$ | **$74.5 \pm 2.0$** | $68.5 \pm 2.3$ |
| RSC-Aug | $72.7 \pm 3.3$ | $67.9 \pm 2.9$ | $65.7 \pm 3.8$ | $60.7 \pm 1.6$ | $66.6 \pm 5.1$ | $66.7 \pm 3.3$ |
| L2D-Aug | $75.9 \pm 0.7$ | $74.6 \pm 0.4$ | $64.0 \pm 0.9$ | $63.7 \pm 2.5$ | $73.6 \pm 2.2$ | $70.4 \pm 1.3$ |
| Ours-Aug | **$77.3 \pm 1.0$** | **$78.2 \pm 2.3$** | **$75.2 \pm 1.8$** | **$75.5 \pm 0.8$** | $73.3 \pm 1.7$ | **$75.9 \pm 1.5$** |

Table 5: Out-of-domain accuracy on Ocelot evaluated only on the organ ENDOMETRIUM. The column name indicates the centre used to train models. The best accuracy for each column is in **bold face** and the second best in *italics*.

| Method | Centre-0 | Centre-1 | Centre-2 | Centre-3 | Centre-4 | Average |
|--------|----------|----------|----------|----------|----------|---------|
| ERM | $68.4 \pm 2.9$ | $58.7 \pm 2.2$ | $52.3 \pm 2.1$ | $47.5 \pm 0.7$ | $53.9 \pm 5.1$ | $56.2 \pm 2.6$ |
| Macenko | $68.9 \pm 2.7$ | $57.4 \pm 2.1$ | $71.6 \pm 2.0$ | $53.7 \pm 1.8$ | $68.4 \pm 3.7$ | $64.0 \pm 2.4$ |
| RSC | $71.3 \pm 2.0$ | $58.1 \pm 1.4$ | $50.8 \pm 0.8$ | $48.4 \pm 1.0$ | $50.5 \pm 1.4$ | $55.8 \pm 1.3$ |
| L2D | $76.6 \pm 3.0$ | $58.0 \pm 0.5$ | $64.5 \pm 2.9$ | $57.4 \pm 4.7$ | $62.1 \pm 4.2$ | $63.7 \pm 3.1$ |
| Ours | $70.6 \pm 1.3$ | **$72.6 \pm 1.4$** | $63.9 \pm 2.2$ | $62.6 \pm 1.4$ | $66.9 \pm 1.5$ | $67.3 \pm 1.6$ |
| ERM-Aug | *$78.5 \pm 1.6$* | $64.1 \pm 2.6$ | $75.1 \pm 2.8$ | $56.5 \pm 2.6$ | $73.0 \pm 3.9$ | $69.4 \pm 2.7$ |
| Macenko-Aug | $73.2 \pm 2.8$ | $64.1 \pm 2.3$ | **$78.7 \pm 2.0$** | $58.5 \pm 4.1$ | $74.5 \pm 2.5$ | $69.8 \pm 2.7$ |
| RSC-Aug | $76.2 \pm 2.7$ | $64.8 \pm 3.8$ | *$75.8 \pm 2.6$* | $57.4 \pm 1.1$ | *$74.7 \pm 5.6$* | $69.8 \pm 3.2$ |
| L2D-Aug | **$78.5 \pm 0.6$** | *$70.6 \pm 0.7$* | $72.0 \pm 0.8$ | *$65.1 \pm 3.5$* | **$79.9 \pm 1.7$** | **$73.2 \pm 1.5$** |
| Ours-Aug | $73.5 \pm 1.8$ | $69.5 \pm 3.0$ | $68.8 \pm 2.4$ | **$71.4 \pm 1.3$** | $69.9 \pm 1.9$ | *$70.6 \pm 2.1$* |

Table 6: Out-of-domain accuracy on Ocelot evaluated only on the organ HEAD AND NECK. The column name indicates the centre used to train models. The best accuracy for each column is in **bold face** and the second best in *italics*.

| Method | Centre-0 | Centre-1 | Centre-2 | Centre-3 | Centre-4 | Average |
|--------|----------|----------|----------|----------|----------|---------|
| ERM | $61.9 \pm 4.0$ | $53.7 \pm 1.6$ | $49.5 \pm 1.8$ | $44.1 \pm 1.5$ | $52.6 \pm 2.9$ | $52.4 \pm 2.4$ |
| Macenko | $57.7 \pm 2.1$ | $54.1 \pm 1.3$ | $51.8 \pm 1.2$ | $56.0 \pm 2.9$ | $61.2 \pm 3.0$ | $56.2 \pm 2.1$ |
| RSC | $63.0 \pm 4.8$ | $55.1 \pm 1.5$ | $50.1 \pm 0.5$ | $44.9 \pm 2.1$ | $50.6 \pm 1.2$ | $52.7 \pm 2.0$ |
| L2D | $63.1 \pm 2.7$ | $59.3 \pm 1.2$ | $47.7 \pm 1.4$ | $57.4 \pm 1.7$ | $60.0 \pm 6.3$ | $57.5 \pm 2.7$ |
| Ours | $66.2 \pm 3.1$ | *$74.1 \pm 1.3$* | *$64.6 \pm 1.0$* | $59.6 \pm 1.4$ | *$69.0 \pm 1.0$* | $66.7 \pm 1.5$ |
| ERM-Aug | *$72.0 \pm 3.8$* | $67.4 \pm 2.4$ | $60.6 \pm 2.1$ | $58.9 \pm 1.8$ | $63.2 \pm 3.9$ | $64.4 \pm 2.8$ |
| Macenko-Aug | $67.2 \pm 1.8$ | $63.3 \pm 1.9$ | $60.4 \pm 3.0$ | $57.6 \pm 3.6$ | $68.7 \pm 2.7$ | $63.4 \pm 2.6$ |
| RSC-Aug | $68.9 \pm 2.1$ | $64.3 \pm 2.4$ | $61.9 \pm 3.7$ | $61.2 \pm 2.6$ | $64.7 \pm 3.9$ | $64.2 \pm 2.9$ |
| L2D-Aug | **$73.7 \pm 1.7$** | $72.7 \pm 0.5$ | $60.2 \pm 1.1$ | *$64.4 \pm 1.9$* | $68.6 \pm 2.3$ | *$67.9 \pm 1.5$* |
| Ours-Aug | $71.8 \pm 1.2$ | **$74.9 \pm 1.8$** | **$68.8 \pm 1.3$** | **$74.8 \pm 1.0$** | $70.4 \pm 0.9$ | **$72.1 \pm 1.3$** |

Table 7: Out-of-domain accuracy on Ocelot evaluated only on the organ KIDNEY. The column name indicates the centre used to train models. The best accuracy for each column is in **bold face** and the second best in *italics*.

| Method | Centre-0 | Centre-1 | Centre-2 | Centre-3 | Centre-4 | Average |
|---|---|---|---|---|---|---|
| ERM | $61.5 \pm 4.2$ | $50.2 \pm 0.4$ | $50.3 \pm 0.5$ | $42.5 \pm 2.7$ | $54.7 \pm 6.2$ | $51.8 \pm 2.8$ |
| Macenko | $58.3 \pm 1.9$ | $50.0 \pm 0.7$ | $58.0 \pm 2.0$ | $52.6 \pm 2.1$ | $62.1 \pm 4.3$ | $56.2 \pm 2.2$ |
| RSC | $62.1 \pm 4.5$ | $50.5 \pm 0.7$ | $50.0 \pm 0.1$ | $45.4 \pm 2.5$ | $50.9 \pm 1.4$ | $51.8 \pm 1.8$ |
| L2D | $54.4 \pm 1.4$ | $53.2 \pm 0.6$ | $47.0 \pm 3.4$ | $51.6 \pm 2.3$ | $57.2 \pm 5.6$ | $52.7 \pm 2.7$ |
| Ours | $54.9 \pm 1.1$ | *$59.5 \pm 2.0$* | $56.5 \pm 3.3$ | $53.9 \pm 1.5$ | $62.7 \pm 2.3$ | $57.5 \pm 2.0$ |
| ERM-Aug | *$66.4 \pm 2.1$* | $54.7 \pm 1.3$ | $61.9 \pm 3.0$ | $52.6 \pm 2.2$ | $65.4 \pm 3.5$ | $60.2 \pm 2.4$ |
| Macenko-Aug | $59.7 \pm 2.0$ | $52.5 \pm 0.9$ | *$63.4 \pm 2.4$* | $54.7 \pm 2.9$ | *$69.0 \pm 1.8$* | $59.9 \pm 2.0$ |
| RSC-Aug | $66.0 \pm 3.4$ | $51.2 \pm 3.1$ | **$65.4 \pm 4.4$** | $56.0 \pm 2.1$ | $68.7 \pm 3.1$ | *$61.5 \pm 3.2$* |
| L2D-Aug | **$69.5 \pm 0.6$** | $58.6 \pm 0.7$ | $59.0 \pm 0.9$ | *$57.7 \pm 3.4$* | **$71.7 \pm 0.6$** | **$63.3 \pm 1.2$** |
| Ours-Aug | $58.1 \pm 1.1$ | **$65.7 \pm 3.2$** | $57.6 \pm 2.4$ | **$59.1 \pm 1.1$** | $64.0 \pm 2.1$ | $60.9 \pm 2.0$ |

Table 8: Out-of-domain accuracy on Ocelot evaluated only on the organ PROSTATE. The column name indicates the centre used to train models. The best accuracy for each column is in **bold face** and the second best in *italics*.

| Method | Centre-0 | Centre-1 | Centre-2 | Centre-3 | Centre-4 | Average |
|---|---|---|---|---|---|---|
| ERM | $68.4 \pm 1.9$ | $58.7 \pm 1.7$ | $57.6 \pm 3.0$ | $46.7 \pm 1.0$ | $52.1 \pm 2.5$ | $56.7 \pm 2.0$ |
| Macenko | $63.2 \pm 2.8$ | $51.5 \pm 0.6$ | $66.8 \pm 2.3$ | $54.0 \pm 1.2$ | $68.5 \pm 4.3$ | $60.8 \pm 2.2$ |
| RSC | $70.8 \pm 2.6$ | $60.9 \pm 1.1$ | $54.9 \pm 2.4$ | $48.0 \pm 1.2$ | $49.9 \pm 0.7$ | $56.9 \pm 1.6$ |
| L2D | $73.1 \pm 0.8$ | $54.2 \pm 0.7$ | $60.9 \pm 2.7$ | $55.6 \pm 2.2$ | $58.9 \pm 2.0$ | $60.5 \pm 1.7$ |
| Ours | $75.4 \pm 0.4$ | **$75.1 \pm 1.2$** | *$75.4 \pm 0.3$* | $68.2 \pm 2.9$ | *$74.7 \pm 0.6$* | *$73.8 \pm 1.1$* |
| ERM-Aug | **$76.0 \pm 1.1$** | $60.7 \pm 2.6$ | $74.2 \pm 1.4$ | $60.3 \pm 1.1$ | $67.8 \pm 2.7$ | $67.8 \pm 1.8$ |
| Macenko-Aug | $72.9 \pm 1.5$ | $58.6 \pm 0.9$ | $74.2 \pm 1.0$ | $60.9 \pm 2.5$ | $74.4 \pm 1.2$ | $68.2 \pm 1.4$ |
| RSC-Aug | *$75.7 \pm 1.7$* | $60.2 \pm 2.3$ | $74.7 \pm 1.6$ | $62.0 \pm 1.1$ | $70.3 \pm 3.1$ | $68.6 \pm 2.0$ |
| L2D-Aug | $75.5 \pm 0.5$ | $67.8 \pm 0.5$ | $72.1 \pm 0.4$ | $64.7 \pm 1.9$ | $74.1 \pm 1.3$ | $70.9 \pm 0.9$ |
| Ours-Aug | $75.0 \pm 0.2$ | *$73.7 \pm 2.1$* | **$76.3 \pm 0.5$** | **$75.9 \pm 0.4$** | **$77.0 \pm 0.5$** | **$75.6 \pm 0.7$** |

Table 9: Out-of-domain accuracy on Ocelot evaluated only on the organ STOMACH. The column name indicates the centre used to train models. The best accuracy for each column is in **bold face** and the second best in *italics*.

| Method | Centre-0 | Centre-1 | Centre-2 | Centre-3 | Centre-4 | Average |
|---|---|---|---|---|---|---|
| ERM | $71.7 \pm 2.4$ | $56.9 \pm 3.3$ | $50.8 \pm 1.6$ | $47.5 \pm 0.7$ | $53.0 \pm 6.4$ | $56.0 \pm 2.9$ |
| Macenko | $61.6 \pm 3.5$ | $54.2 \pm 1.9$ | $62.9 \pm 3.2$ | $51.4 \pm 3.1$ | $65.5 \pm 3.6$ | $59.1 \pm 3.1$ |
| RSC | $72.1 \pm 1.8$ | $52.1 \pm 0.7$ | $50.1 \pm 0.6$ | $48.6 \pm 0.9$ | $49.6 \pm 1.7$ | $54.5 \pm 1.1$ |
| L2D | $74.1 \pm 1.4$ | $63.9 \pm 1.2$ | $57.9 \pm 1.9$ | $57.9 \pm 4.5$ | $68.0 \pm 3.8$ | $64.4 \pm 2.5$ |
| Ours | $74.4 \pm 1.4$ | **$77.2 \pm 0.7$** | *$73.4 \pm 0.7$* | *$70.9 \pm 1.6$* | $73.9 \pm 1.0$ | *$74.0 \pm 1.1$* |
| ERM-Aug | $76.6 \pm 1.5$ | $68.0 \pm 2.0$ | $70.6 \pm 2.2$ | $53.8 \pm 1.6$ | $72.2 \pm 2.4$ | $68.2 \pm 2.0$ |
| Macenko-Aug | $70.9 \pm 3.1$ | $65.8 \pm 2.7$ | $72.4 \pm 1.9$ | $54.3 \pm 2.4$ | $73.5 \pm 2.0$ | $67.4 \pm 2.4$ |
| RSC-Aug | $73.3 \pm 2.8$ | $68.5 \pm 2.5$ | $70.6 \pm 1.8$ | $55.3 \pm 2.0$ | $74.7 \pm 2.0$ | $68.5 \pm 2.2$ |
| L2D-Aug | *$77.0 \pm 0.5$* | $72.0 \pm 0.6$ | $67.1 \pm 1.0$ | $61.7 \pm 2.9$ | *$76.4 \pm 0.7$* | $70.8 \pm 1.1$ |
| Ours-Aug | **$77.5 \pm 0.7$** | *$76.5 \pm 3.5$* | **$75.9 \pm 1.0$** | **$74.6 \pm 0.8$** | **$76.4 \pm 0.9$** | **$76.2 \pm 1.4$** |

Table 10: Out-of-domain accuracy on CAMELYON17 where embeddings for segmentation masks were subtracted from the embeddings of the original image to see if the accuracy drops. The column name indicates the centre used to train models. The best accuracy for each column is in **bold face** and the second best in *italics*.

| Method | Centre-0 | Centre-1 | Centre-2 | Centre-3 | Centre-4 | Average |
|---|---|---|---|---|---|---|
| ERM | $63.0 \pm 7.4$ | $64.2 \pm 8.3$ | $58.2 \pm 3.6$ | $52.3 \pm 3.1$ | $52.6 \pm 3.1$ | $58.1 \pm 5.1$ |
| Macenko | $63.3 \pm 13.2$ | $58.4 \pm 11.5$ | $54.5 \pm 3.9$ | $59.8 \pm 2.8$ | $52.1 \pm 5.5$ | $57.6 \pm 7.4$ |
| RSC | $54.6 \pm 3.5$ | $52.7 \pm 3.1$ | $64.0 \pm 7.6$ | $51.5 \pm 5.0$ | $54.1 \pm 6.9$ | $55.4 \pm 5.2$ |
| L2D | $\mathbf{86.4 \pm 1.9}$ | $48.7 \pm 5.0$ | $70.1 \pm 8.2$ | $\mathbf{71.5 \pm 4.8}$ | $\mathbf{73.4 \pm 7.9}$ | $\mathit{70.0 \pm 5.6}$ |
| Ours | $36.6 \pm 4.6$ | $51.4 \pm 6.5$ | $38.3 \pm 4.9$ | $37.5 \pm 5.2$ | $56.6 \pm 8.8$ | $44.1 \pm 6.0$ |
| ERM-Aug | $75.7 \pm 13.3$ | $61.2 \pm 9.2$ | $72.1 \pm 12.5$ | $65.1 \pm 10.2$ | $64.8 \pm 12.3$ | $67.8 \pm 11.5$ |
| Macenko-Aug | $76.0 \pm 11.8$ | $\mathbf{68.7 \pm 8.0}$ | $60.1 \pm 7.1$ | $62.7 \pm 8.5$ | $62.9 \pm 10.5$ | $66.1 \pm 9.2$ |
| RSC-Aug | $69.4 \pm 9.0$ | $64.4 \pm 8.4$ | $\mathit{82.1 \pm 7.9}$ | $56.0 \pm 5.9$ | $\mathit{68.5 \pm 11.4}$ | $68.1 \pm 8.5$ |
| L2D-Aug | $\mathit{82.2 \pm 2.7}$ | $\mathit{67.3 \pm 2.1}$ | $\mathbf{82.2 \pm 3.5}$ | $\mathit{70.4 \pm 6.9}$ | $61.7 \pm 4.2$ | $\mathbf{72.8 \pm 3.9}$ |
| Ours-Aug | $47.3 \pm 6.6$ | $41.3 \pm 4.3$ | $45.4 \pm 6.5$ | $39.6 \pm 3.9$ | $48.4 \pm 2.9$ | $44.4 \pm 4.8$ |

Table 11: Out-of-domain accuracy on CAMELYON17 where nuclei are blackened out. The column name indicates the centre used to train models. The best accuracy for each column is in **bold face** and the second best in *italics*.

| Method | Centre-0 | Centre-1 | Centre-2 | Centre-3 | Centre-4 | Average |
|---|---|---|---|---|---|---|
| ERM | $61.2 \pm 7.3$ | $53.1 \pm 2.7$ | $55.7 \pm 11.7$ | $50.1 \pm 0.1$ | $51.2 \pm 3.6$ | $54.3 \pm 5.1$ |
| Macenko | $50.3 \pm 0.9$ | $57.5 \pm 12.6$ | $50.5 \pm 1.3$ | $56.1 \pm 7.0$ | $51.7 \pm 3.3$ | $53.2 \pm 5.0$ |
| RSC | $62.5 \pm 6.5$ | $52.8 \pm 2.4$ | $52.0 \pm 5.9$ | $50.4 \pm 0.8$ | $49.1 \pm 3.5$ | $53.4 \pm 3.8$ |
| L2D | $83.7 \pm 4.6$ | $89.4 \pm 3.4$ | $83.1 \pm 7.5$ | $70.6 \pm 8.8$ | $63.7 \pm 12.1$ | $78.1 \pm 7.3$ |
| Ours | $91.2 \pm 1.0$ | $\mathbf{92.4 \pm 0.3}$ | $\mathit{91.6 \pm 1.4}$ | $\mathit{89.1 \pm 2.1}$ | $\mathit{86.0 \pm 1.6}$ | $\mathit{90.1 \pm 1.3}$ |
| ERM-Aug | $79.4 \pm 8.7$ | $57.6 \pm 8.3$ | $85.8 \pm 4.4$ | $58.4 \pm 8.1$ | $50.4 \pm 0.8$ | $66.3 \pm 6.1$ |
| Macenko-Aug | $75.6 \pm 11.6$ | $87.9 \pm 6.2$ | $64.2 \pm 12.4$ | $59.8 \pm 8.4$ | $71.5 \pm 12.7$ | $71.8 \pm 10.3$ |
| RSC-Aug | $74.5 \pm 10.7$ | $63.2 \pm 13.6$ | $86.8 \pm 7.9$ | $57.3 \pm 5.7$ | $50.7 \pm 0.8$ | $66.5 \pm 7.7$ |
| L2D-Aug | $\mathit{91.9 \pm 0.3}$ | $87.3 \pm 0.8$ | $77.0 \pm 3.3$ | $87.7 \pm 2.6$ | $79.5 \pm 7.8$ | $84.7 \pm 3.0$ |
| Ours-Aug | $\mathbf{92.3 \pm 0.5}$ | $\mathit{91.7 \pm 1.4}$ | $\mathbf{92.1 \pm 1.2}$ | $\mathbf{93.4 \pm 0.3}$ | $\mathbf{86.2 \pm 3.2}$ | $\mathbf{91.2 \pm 1.3}$ |

Table 12: Out-of-domain accuracy on CAMELYON17 where nuclei are blackened out after being expanded with filter size 5, i.e., the blackened out part covers nuclei and some region around nuclei. The column name indicates the centre used to train models. The best accuracy for each column is in **bold face** and the second best in *italics*.

| Method | Centre-0 | Centre-1 | Centre-2 | Centre-3 | Centre-4 | Average |
|---|---|---|---|---|---|---|
| ERM | $56.7 \pm 6.1$ | $52.4 \pm 2.7$ | $55.7 \pm 12.2$ | $50.1 \pm 0.2$ | $50.0 \pm 3.3$ | $53.0 \pm 4.9$ |
| Macenko | $50.3 \pm 1.1$ | $54.8 \pm 9.3$ | $50.0 \pm 0.1$ | $53.7 \pm 7.2$ | $50.2 \pm 0.4$ | $51.8 \pm 3.6$ |
| RSC | $57.2 \pm 5.9$ | $51.1 \pm 2.4$ | $50.8 \pm 2.3$ | $50.3 \pm 0.5$ | $49.7 \pm 0.6$ | $51.8 \pm 2.4$ |
| L2D | $78.0 \pm 8.4$ | $\mathbf{86.0 \pm 8.7}$ | $83.1 \pm 8.3$ | $62.4 \pm 9.0$ | $62.3 \pm 13.0$ | $74.4 \pm 9.5$ |
| Ours | $89.9 \pm 0.4$ | $\mathit{80.3 \pm 3.5}$ | $\mathit{90.9 \pm 1.5}$ | $\mathit{90.7 \pm 1.0}$ | $\mathit{86.3 \pm 2.0}$ | $\mathbf{87.6 \pm 1.7}$ |
| ERM-Aug | $72.3 \pm 11.2$ | $51.6 \pm 2.0$ | $78.6 \pm 10.0$ | $50.9 \pm 2.3$ | $50.1 \pm 0.2$ | $60.7 \pm 5.1$ |
| Macenko-Aug | $68.7 \pm 12.3$ | $79.7 \pm 11.5$ | $57.8 \pm 10.1$ | $53.2 \pm 7.3$ | $64.7 \pm 13.1$ | $64.8 \pm 10.9$ |
| RSC-Aug | $69.5 \pm 11.8$ | $59.8 \pm 12.4$ | $85.7 \pm 7.0$ | $50.2 \pm 0.5$ | $50.1 \pm 0.2$ | $63.1 \pm 6.4$ |
| L2D-Aug | $\mathbf{90.3 \pm 0.6}$ | $78.6 \pm 2.3$ | $67.5 \pm 6.1$ | $81.7 \pm 5.5$ | $78.8 \pm 10.1$ | $79.4 \pm 4.9$ |
| Ours-Aug | $\mathit{89.9 \pm 0.4}$ | $75.0 \pm 2.6$ | $\mathbf{91.4 \pm 1.1}$ | $\mathbf{91.5 \pm 0.4}$ | $\mathbf{86.3 \pm 3.6}$ | $\mathit{86.8 \pm 1.6}$ |

Table 13: Out-of-domain accuracy on CAMELYON17 where nuclei are blackened out after being expanded with filter size 9, i.e., the blackened out part covers nuclei and some region around nuclei. The column name indicates the centre used to train models. The best accuracy for each column is in **bold face** and the second best in *italics*.

| Method | Centre-0 | Centre-1 | Centre-2 | Centre-3 | Centre-4 | Average |
|---|---|---|---|---|---|---|
| ERM | $54.3 \pm 4.9$ | $51.9 \pm 2.5$ | $55.7 \pm 12.2$ | $50.1 \pm 0.2$ | $51.1 \pm 3.2$ | $52.6 \pm 4.6$ |
| Macenko | $50.3 \pm 0.9$ | $53.4 \pm 5.8$ | $50.1 \pm 0.1$ | $54.5 \pm 7.7$ | $50.1 \pm 0.1$ | $51.7 \pm 2.9$ |
| RSC | $54.0 \pm 4.6$ | $50.4 \pm 2.3$ | $50.3 \pm 1.0$ | $50.2 \pm 0.4$ | $49.8 \pm 0.4$ | $50.9 \pm 1.7$ |
| L2D | $70.9 \pm 10.9$ | $\mathbf{81.5 \pm 10.6}$ | *76.0 ± 12.3* | $58.1 \pm 8.6$ | $57.8 \pm 9.7$ | $68.9 \pm 10.4$ |
| Ours | *76.6 ± 3.3* | $67.0 \pm 5.0$ | $66.9 \pm 5.0$ | $\mathbf{88.6 \pm 0.8}$ | $\mathbf{83.0 \pm 1.6}$ | $\mathbf{76.4 \pm 3.2}$ |
| ERM-Aug | $66.6 \pm 11.3$ | $50.1 \pm 0.3$ | $71.7 \pm 11.4$ | $50.2 \pm 0.4$ | $50.0 \pm 0.2$ | $57.7 \pm 4.7$ |
| Macenko-Aug | $61.9 \pm 10.8$ | $71.6 \pm 10.6$ | $54.9 \pm 8.1$ | $51.6 \pm 4.7$ | $58.7 \pm 9.2$ | $59.7 \pm 8.7$ |
| RSC-Aug | $65.3 \pm 11.8$ | $56.3 \pm 9.7$ | $\mathbf{77.7 \pm 9.5}$ | $50.1 \pm 0.0$ | $50.1 \pm 0.1$ | $59.9 \pm 6.2$ |
| L2D-Aug | $\mathbf{87.1 \pm 0.5}$ | *71.8 ± 4.5* | $54.3 \pm 3.0$ | $71.9 \pm 8.3$ | $76.3 \pm 11.3$ | $72.3 \pm 5.5$ |
| Ours-Aug | $75.1 \pm 1.3$ | $62.1 \pm 4.7$ | $69.7 \pm 3.9$ | *85.1 ± 2.2* | *80.4 ± 5.3* | *74.5 ± 3.5* |

Table 14: Out-of-domain accuracy on CAMELYON17 where nuclei are whitened out. The column name indicates the centre used to train models. The best accuracy for each column is in **bold face** and the second best in *italics*.

| Method | Centre-0 | Centre-1 | Centre-2 | Centre-3 | Centre-4 | Average |
|---|---|---|---|---|---|---|
| ERM | $61.3 \pm 2.8$ | $50.0 \pm 1.4$ | $50.0 \pm 0.0$ | $50.1 \pm 0.1$ | $48.3 \pm 2.4$ | $51.9 \pm 1.3$ |
| Macenko | $50.2 \pm 0.5$ | $50.0 \pm 0.0$ | $50.0 \pm 0.1$ | $50.5 \pm 0.9$ | $50.0 \pm 0.1$ | $50.1 \pm 0.3$ |
| RSC | $65.6 \pm 3.8$ | $50.7 \pm 1.8$ | $50.0 \pm 0.0$ | $50.3 \pm 0.3$ | $50.4 \pm 1.4$ | $53.4 \pm 1.5$ |
| L2D | $72.9 \pm 9.7$ | $71.1 \pm 6.1$ | $54.3 \pm 8.1$ | *72.9 ± 4.1* | $53.3 \pm 4.1$ | $64.9 \pm 6.4$ |
| Ours | *79.6 ± 6.0* | *87.8 ± 2.1* | $\mathbf{72.1 \pm 6.5}$ | $\mathbf{75.9 \pm 5.2}$ | *60.7 ± 5.2* | $\mathbf{75.2 \pm 5.0}$ |
| ERM-Aug | $51.3 \pm 2.8$ | $46.6 \pm 6.2$ | $64.4 \pm 11.0$ | $49.1 \pm 2.6$ | $50.9 \pm 2.5$ | $52.5 \pm 5.0$ |
| Macenko-Aug | $49.6 \pm 1.2$ | $48.7 \pm 4.5$ | $55.7 \pm 10.5$ | $50.4 \pm 0.5$ | $50.0 \pm 0.0$ | $50.9 \pm 3.3$ |
| RSC-Aug | $52.9 \pm 4.5$ | $50.2 \pm 0.5$ | $52.7 \pm 2.8$ | $50.0 \pm 0.0$ | $53.5 \pm 6.1$ | $51.9 \pm 2.8$ |
| L2D-Aug | $\mathbf{85.5 \pm 1.1}$ | $55.8 \pm 3.8$ | $50.5 \pm 0.2$ | $59.4 \pm 7.1$ | $54.5 \pm 4.5$ | $61.1 \pm 3.3$ |
| Ours-Aug | $67.4 \pm 7.6$ | $\mathbf{92.1 \pm 0.6}$ | *67.3 ± 11.8* | $59.0 \pm 7.2$ | $\mathbf{85.5 \pm 4.7}$ | *74.3 ± 6.4* |

Table 15: Out-of-domain accuracy on CAMELYON17 where nuclei are whitened out after being expanded with filter size 5, i.e., the whitened out part covers nuclei and some region around nuclei. The column name indicates the centre used to train models. The best accuracy for each column is in **bold face** and the second best in *italics*.

| Method | Centre-0 | Centre-1 | Centre-2 | Centre-3 | Centre-4 | Average |
|---|---|---|---|---|---|---|
| ERM | $61.3 \pm 3.0$ | $50.7 \pm 1.8$ | $50.0 \pm 0.0$ | $50.1 \pm 0.3$ | $49.6 \pm 1.3$ | $52.3 \pm 1.3$ |
| Macenko | $50.5 \pm 0.9$ | $50.0 \pm 0.0$ | $50.0 \pm 0.1$ | $50.3 \pm 0.6$ | $50.0 \pm 0.1$ | $50.2 \pm 0.3$ |
| RSC | $63.4 \pm 4.8$ | $50.6 \pm 1.7$ | $50.0 \pm 0.0$ | $50.3 \pm 0.3$ | $50.3 \pm 0.8$ | $52.9 \pm 1.5$ |
| L2D | *69.0 ± 7.5* | $64.2 \pm 6.2$ | $53.8 \pm 7.0$ | $\mathbf{70.4 \pm 5.6}$ | $52.5 \pm 8.4$ | $62.0 \pm 6.9$ |
| Ours | $63.9 \pm 9.3$ | *74.6 ± 2.9* | $65.5 \pm 7.1$ | *65.6 ± 5.9* | $51.4 \pm 0.8$ | *64.2 ± 5.2* |
| ERM-Aug | $50.9 \pm 2.6$ | $49.9 \pm 0.2$ | *69.3 ± 9.3* | $50.1 \pm 0.2$ | $51.2 \pm 1.3$ | $54.3 \pm 2.7$ |
| Macenko-Aug | $51.2 \pm 1.3$ | $50.2 \pm 0.3$ | $61.9 \pm 8.5$ | $52.5 \pm 2.6$ | $50.4 \pm 1.4$ | $53.2 \pm 2.8$ |
| RSC-Aug | $52.6 \pm 4.0$ | $50.4 \pm 0.8$ | $55.8 \pm 6.2$ | $50.1 \pm 0.0$ | $54.6 \pm 6.6$ | $52.7 \pm 3.5$ |
| L2D-Aug | $\mathbf{80.4 \pm 3.2}$ | $51.0 \pm 1.3$ | $50.5 \pm 0.2$ | $59.4 \pm 6.8$ | *61.4 ± 7.6* | $60.5 \pm 3.8$ |
| Ours-Aug | $53.8 \pm 3.2$ | $\mathbf{82.9 \pm 2.9}$ | $\mathbf{71.2 \pm 11.4}$ | $55.8 \pm 6.0$ | $\mathbf{71.2 \pm 9.9}$ | $\mathbf{67.0 \pm 6.7}$ |

Table 16: Out-of-domain accuracy on CAMELYON17 where nuclei are whitened out after being expanded with filter size 9, i.e., the whitened out part covers nuclei and some region around nuclei. The column name indicates the centre used to train models. The best accuracy for each column is in **bold face** and the second best in *italics*.

| Method | Centre-0 | Centre-1 | Centre-2 | Centre-3 | Centre-4 | Average |
|---|---|---|---|---|---|---|
| ERM | $61.8 \pm 3.5$ | $50.5 \pm 1.6$ | $50.0 \pm 0.0$ | $50.1 \pm 0.2$ | $49.9 \pm 1.2$ | $52.5 \pm 1.3$ |
| Macenko | $50.5 \pm 0.9$ | $50.0 \pm 0.0$ | $50.0 \pm 0.1$ | $50.1 \pm 0.3$ | $50.0 \pm 0.2$ | $50.1 \pm 0.3$ |
| RSC | $59.6 \pm 5.0$ | $50.2 \pm 1.5$ | $50.0 \pm 0.0$ | $50.3 \pm 0.3$ | $50.1 \pm 0.5$ | $52.0 \pm 1.5$ |
| L2D | *$63.5 \pm 4.9$* | $55.7 \pm 3.9$ | $53.0 \pm 5.2$ | **$63.9 \pm 5.0$** | $48.0 \pm 9.6$ | $56.8 \pm 5.7$ |
| Ours | $55.9 \pm 6.9$ | *$67.3 \pm 3.1$* | $61.4 \pm 6.8$ | *$58.1 \pm 4.7$* | $50.9 \pm 0.7$ | *$58.7 \pm 4.4$* |
| ERM-Aug | $51.0 \pm 1.6$ | $50.0 \pm 0.0$ | **$68.5 \pm 10.5$** | $50.2 \pm 0.3$ | $47.3 \pm 3.2$ | $53.4 \pm 3.1$ |
| Macenko-Aug | $51.3 \pm 1.0$ | $50.0 \pm 0.1$ | $62.5 \pm 9.4$ | $51.8 \pm 1.8$ | $47.7 \pm 3.0$ | $52.7 \pm 3.1$ |
| RSC-Aug | $51.3 \pm 2.0$ | $50.1 \pm 0.3$ | $53.8 \pm 6.2$ | $50.1 \pm 0.0$ | $50.4 \pm 4.4$ | $51.1 \pm 2.6$ |
| L2D-Aug | **$68.2 \pm 4.0$** | $49.7 \pm 0.1$ | $50.2 \pm 0.1$ | $58.0 \pm 5.7$ | *$51.6 \pm 8.1$* | $55.5 \pm 3.6$ |
| Ours-Aug | $50.3 \pm 0.3$ | **$76.9 \pm 4.3$** | *$67.6 \pm 9.7$* | $53.1 \pm 3.9$ | **$58.9 \pm 8.1$** | **$61.3 \pm 5.3$** |

Table 17: Out-of-domain accuracy on where nuclei are *inpainted* after being expanded with filter size 5. The column name indicates the centre used to train models. The best accuracy for each column is in **bold face** and the second best in *italics*.

| Method | Centre-0 | Centre-1 | Centre-2 | Centre-3 | Centre-4 | Average |
|---|---|---|---|---|---|---|
| ERM | $54.7 \pm 3.3$ | *$62.4 \pm 3.7$* | $50.5 \pm 1.9$ | $49.9 \pm 0.2$ | $48.6 \pm 7.1$ | $53.2 \pm 3.2$ |
| Macenko | $54.3 \pm 7.6$ | $54.5 \pm 3.4$ | $50.0 \pm 0.0$ | $49.9 \pm 0.5$ | $49.6 \pm 6.2$ | $51.7 \pm 3.5$ |
| RSC | $59.0 \pm 2.2$ | $56.7 \pm 2.5$ | $49.9 \pm 0.1$ | $50.0 \pm 0.1$ | $51.9 \pm 4.8$ | $53.5 \pm 1.9$ |
| L2D | *$68.0 \pm 3.9$* | **$66.1 \pm 0.9$** | $50.2 \pm 0.4$ | *$61.7 \pm 3.7$* | **$52.9 \pm 5.8$** | *$59.8 \pm 2.9$* |
| Ours | $50.0 \pm 0.0$ | $50.5 \pm 2.1$ | $49.2 \pm 0.6$ | $50.0 \pm 0.0$ | $49.0 \pm 0.9$ | $49.7 \pm 0.7$ |
| ERM-Aug | $51.8 \pm 3.1$ | $59.1 \pm 4.8$ | *$57.5 \pm 5.9$* | $58.5 \pm 3.1$ | $46.1 \pm 3.1$ | $54.6 \pm 4.0$ |
| Macenko-Aug | $55.5 \pm 7.5$ | $54.8 \pm 3.8$ | **$61.9 \pm 4.3$** | $55.3 \pm 3.9$ | $52.0 \pm 8.5$ | $55.9 \pm 5.6$ |
| RSC-Aug | $48.6 \pm 3.5$ | $60.0 \pm 5.6$ | $56.0 \pm 6.2$ | $60.2 \pm 2.4$ | $43.2 \pm 5.7$ | $53.6 \pm 4.7$ |
| L2D-Aug | **$69.9 \pm 1.4$** | $60.0 \pm 1.0$ | $52.1 \pm 1.5$ | **$67.9 \pm 2.2$** | *$52.5 \pm 8.8$* | **$60.5 \pm 3.0$** |
| Ours-Aug | $50.0 \pm 0.0$ | $48.9 \pm 1.6$ | $50.3 \pm 1.4$ | $50.0 \pm 0.0$ | $49.8 \pm 0.1$ | $49.8 \pm 0.6$ |

Table 18: In domain accuracy on CAMELYON17 where nuclei are *inpainted* after being expanded with filter size 5. The column name indicates the centre used to train models. The best accuracy for each column is in **bold face** and the second best in *italics*.

| Method | Centre-0 | Centre-1 | Centre-2 | Centre-3 | Centre-4 | Average |
|---|---|---|---|---|---|---|
| ERM | $58.1 \pm 4.0$ | *$63.6 \pm 8.4$* | $47.9 \pm 0.3$ | $48.8 \pm 0.2$ | $45.2 \pm 5.8$ | $52.7 \pm 3.7$ |
| Macenko | $53.4 \pm 9.3$ | $53.7 \pm 0.2$ | $48.2 \pm 0.0$ | $48.3 \pm 0.5$ | $52.5 \pm 4.3$ | $51.2 \pm 2.8$ |
| RSC | $58.2 \pm 4.8$ | $59.5 \pm 12.4$ | $48.1 \pm 0.1$ | $48.5 \pm 0.1$ | $49.9 \pm 6.4$ | $52.8 \pm 4.8$ |
| L2D | $66.1 \pm 8.5$ | **$70.9 \pm 2.5$** | $48.1 \pm 0.1$ | $53.6 \pm 5.2$ | $51.4 \pm 11.3$ | $58.0 \pm 5.5$ |
| Ours | $49.6 \pm 0.2$ | $53.5 \pm 0.7$ | $47.9 \pm 0.4$ | $48.3 \pm 0.0$ | $45.2 \pm 1.0$ | $48.9 \pm 0.5$ |
| ERM-Aug | $56.3 \pm 6.8$ | $57.2 \pm 3.0$ | $62.6 \pm 11.5$ | $82.5 \pm 5.4$ | $44.6 \pm 3.3$ | $60.6 \pm 6.0$ |
| Macenko-Aug | *$66.2 \pm 5.3$* | $54.4 \pm 1.3$ | $66.4 \pm 14.4$ | $59.4 \pm 6.8$ | *$59.0 \pm 10.2$* | $61.1 \pm 7.6$ |
| RSC-Aug | $65.4 \pm 8.8$ | $58.3 \pm 10.6$ | **$72.7 \pm 8.6$** | *$88.2 \pm 2.6$* | $44.5 \pm 3.2$ | *$65.8 \pm 6.8$* |
| L2D-Aug | **$72.8 \pm 0.7$** | $37.6 \pm 2.2$ | *$68.2 \pm 3.1$* | **$88.9 \pm 1.3$** | **$62.5 \pm 6.6$** | **$66.0 \pm 2.8$** |
| Ours-Aug | $49.9 \pm 0.0$ | $52.3 \pm 2.7$ | $48.0 \pm 0.1$ | $48.1 \pm 0.2$ | $45.0 \pm 1.1$ | $48.6 \pm 0.8$ |

Table 19: OUT-OF-DOMAIN recall on CAMELYON17 where nuclei are *inpainted* after being expanded with filter size 5. The column name indicates the centre used to train models. The best accuracy for each column is in **bold face** and the second best in *italics*.

| Method | Centre-0 | Centre-1 | Centre-2 | Centre-3 | Centre-4 | Average |
|---|---|---|---|---|---|---|
| ERM | **91.4 ± 11.5** | 60.0 ± 8.9 | 1.7 ± 4.6 | 0.1 ± 0.1 | 21.9 ± 20.3 | 35.0 ± 9.1 |
| Macenko | 11.1 ± 21.4 | 12.4 ± 10.3 | 0.0 ± 0.0 | 0.9 ± 2.0 | 20.7 ± 13.6 | 9.0 ± 9.5 |
| RSC | *89.9 ± 8.3* | **67.3 ± 4.3** | 0.0 ± 0.1 | 0.3 ± 0.2 | 21.8 ± 17.2 | *35.9 ± 6.0* |
| L2D | 72.8 ± 19.8 | 44.3 ± 2.8 | 1.0 ± 1.6 | *32.6 ± 12.9* | 19.9 ± 22.5 | 34.1 ± 11.9 |
| Ours | 0.0 ± 0.0 | 2.3 ± 7.0 | 0.2 ± 0.2 | 0.0 ± 0.0 | 0.3 ± 0.2 | 0.6 ± 1.5 |
| ERM-Aug | 10.1 ± 11.2 | 23.8 ± 9.0 | 19.6 ± 17.5 | 24.8 ± 9.7 | 4.8 ± 3.8 | 16.6 ± 10.2 |
| Macenko-Aug | 46.6 ± 21.4 | 13.4 ± 8.9 | **31.4 ± 19.9** | 11.2 ± 8.3 | **44.0 ± 21.1** | 29.3 ± 15.9 |
| RSC-Aug | 23.4 ± 16.5 | 30.4 ± 14.5 | *20.6 ± 12.0* | 22.3 ± 5.8 | 3.1 ± 2.8 | 20.0 ± 10.3 |
| L2D-Aug | 74.5 ± 3.3 | *61.7 ± 4.7* | 12.5 ± 2.6 | **43.6 ± 7.5** | *36.2 ± 7.8* | **45.7 ± 5.2** |
| Ours-Aug | 0.1 ± 0.0 | 2.2 ± 2.8 | 2.0 ± 3.7 | 0.0 ± 0.0 | 0.0 ± 0.0 | 0.8 ± 1.3 |

Table 20: IN DOMAIN recall on CAMELYON17 where nuclei are *inpainted* after being expanded with filter size 5. The column name indicates the centre used to train models. The best accuracy for each column is in **bold face** and the second best in *italics*.

| Method | Centre-0 | Centre-1 | Centre-2 | Centre-3 | Centre-4 | Average |
|---|---|---|---|---|---|---|
| ERM | 97.1 ± 4.3 | 35.0 ± 18.5 | 0.0 ± 0.0 | 1.0 ± 0.3 | 50.6 ± 15.3 | 36.8 ± 7.7 |
| Macenko | 8.7 ± 21.7 | 0.5 ± 0.7 | 0.0 ± 0.0 | 0.9 ± 0.2 | 17.6 ± 11.6 | 5.5 ± 6.8 |
| RSC | **99.4 ± 0.5** | **47.9 ± 25.3** | 0.0 ± 0.0 | 0.4 ± 0.2 | *50.8 ± 17.3* | 39.7 ± 8.7 |
| L2D | 88.5 ± 17.8 | 43.6 ± 7.3 | 0.0 ± 0.0 | 11.3 ± 10.7 | 38.4 ± 29.2 | 36.4 ± 13.0 |
| Ours | 0.1 ± 0.1 | 0.4 ± 1.2 | 0.0 ± 0.0 | 0.0 ± 0.0 | 0.1 ± 0.1 | 0.1 ± 0.3 |
| ERM-Aug | 29.7 ± 26.6 | 9.5 ± 7.3 | 28.8 ± 23.2 | 84.3 ± 13.4 | 5.4 ± 3.8 | 31.6 ± 14.8 |
| Macenko-Aug | 71.9 ± 19.3 | 4.0 ± 2.6 | 37.5 ± 31.0 | 23.7 ± 15.0 | **55.1 ± 24.0** | 38.5 ± 18.4 |
| RSC-Aug | 68.9 ± 31.9 | 17.8 ± 19.8 | **51.2 ± 20.4** | *86.7 ± 5.7* | 3.5 ± 2.5 | *45.6 ± 16.1* |
| L2D-Aug | *99.0 ± 0.1* | *46.7 ± 4.5* | *39.8 ± 6.5* | **93.0 ± 2.0** | 48.7 ± 12.3 | **65.4 ± 5.1** |
| Ours-Aug | 0.0 ± 0.0 | 0.3 ± 0.4 | 0.1 ± 0.1 | 0.1 ± 0.1 | 0.3 ± 0.3 | 0.2 ± 0.2 |

Table 21: OUT-OF-DOMAIN precision on CAMELYON17 where nuclei are *inpainted* after being expanded with filter size 5. The column name indicates the centre used to train models. The best accuracy for each column is in **bold face** and the second best in *italics*.

| Method | Centre-0 | Centre-1 | Centre-2 | Centre-3 | Centre-4 | Average |
|---|---|---|---|---|---|---|
| ERM | 52.7 ± 2.2 | 64.4 ± 6.5 | 39.1 ± 22.0 | 53.4 ± 20.5 | 39.2 ± 21.3 | 49.8 ± 14.5 |
| Macenko | **88.2 ± 8.1** | **83.0 ± 7.6** | 52.8 ± 45.6 | 42.6 ± 19.3 | 48.3 ± 16.9 | 63.0 ± 19.5 |
| RSC | 55.7 ± 2.0 | 55.7 ± 2.4 | 15.7 ± 11.9 | 70.6 ± 17.8 | 50.3 ± 14.6 | 49.6 ± 9.7 |
| L2D | 69.2 ± 9.2 | 78.8 ± 2.6 | 60.7 ± 13.1 | 79.6 ± 5.5 | *54.2 ± 15.3* | *68.5 ± 9.1* |
| Ours | 23.2 ± 10.2 | 20.2 ± 18.3 | 10.2 ± 3.4 | 53.2 ± 21.3 | 11.2 ± 2.0 | 23.6 ± 11.0 |
| ERM-Aug | 61.0 ± 13.0 | *80.5 ± 13.3* | *85.0 ± 8.1* | 76.6 ± 6.5 | 30.8 ± 17.0 | 66.8 ± 11.6 |
| Macenko-Aug | 57.4 ± 11.6 | 77.5 ± 10.2 | **87.8 ± 9.6** | **96.1 ± 2.1** | 50.3 ± 11.2 | **73.8 ± 9.0** |
| RSC-Aug | 47.3 ± 15.2 | 75.4 ± 8.3 | 72.3 ± 19.8 | *92.1 ± 2.6* | 14.9 ± 7.6 | 60.4 ± 10.7 |
| L2D-Aug | 68.3 ± 2.0 | 59.8 ± 1.5 | 62.1 ± 8.1 | 85.4 ± 3.4 | **55.3 ± 13.0** | 66.2 ± 5.6 |
| Ours-Aug | *69.8 ± 10.1* | 34.7 ± 5.6 | 38.3 ± 22.6 | 38.6 ± 16.7 | 2.4 ± 1.2 | 36.8 ± 11.3 |

Table 22: IN DOMAIN precision on CAMELYON17 where nuclei are *inpainted* after being expanded with filter size 5. The column name indicates the centre used to train models. The best accuracy for each column is in **bold face** and the second best in *italics*.

| Method | Centre-0 | Centre-1 | Centre-2 | Centre-3 | Centre-4 | Average |
|---|---|---|---|---|---|---|
| ERM | $54.8 \pm 2.5$ | $71.2 \pm 13.8$ | $0.0 \pm 0.0$ | **$99.4 \pm 0.5$** | $47.7 \pm 7.4$ | $54.6 \pm 4.8$ |
| Macenko | $64.2 \pm 25.8$ | $29.0 \pm 17.0$ | $0.0 \pm 0.0$ | $62.6 \pm 21.6$ | **$75.0 \pm 18.7$** | $46.2 \pm 16.6$ |
| RSC | $54.7 \pm 2.9$ | $56.9 \pm 18.6$ | $1.2 \pm 4.0$ | *$98.4 \pm 1.0$* | $51.3 \pm 9.9$ | $52.5 \pm 7.3$ |
| L2D | $64.0 \pm 10.9$ | **$87.0 \pm 2.6$** | $0.8 \pm 1.8$ | $90.1 \pm 5.9$ | $48.1 \pm 15.3$ | $58.0 \pm 7.3$ |
| Ours | $11.1 \pm 5.8$ | $14.2 \pm 21.4$ | $1.6 \pm 2.6$ | $42.9 \pm 47.5$ | $0.8 \pm 0.8$ | $14.1 \pm 15.6$ |
| ERM-Aug | $58.8 \pm 12.9$ | *$79.8 \pm 12.3$* | **$97.6 \pm 1.6$** | $83.0 \pm 5.6$ | $38.1 \pm 21.3$ | *$71.5 \pm 10.7$* |
| Macenko-Aug | *$64.7 \pm 2.8$* | $61.6 \pm 14.2$ | $94.7 \pm 4.7$ | $92.5 \pm 3.0$ | $60.3 \pm 10.5$ | **$74.8 \pm 7.0$** |
| RSC-Aug | $61.9 \pm 7.6$ | $69.9 \pm 20.5$ | $94.7 \pm 5.4$ | $90.3 \pm 3.8$ | $27.6 \pm 18.1$ | $68.9 \pm 11.1$ |
| L2D-Aug | **$65.1 \pm 0.6$** | $36.3 \pm 1.2$ | *$97.3 \pm 1.3$* | $86.6 \pm 2.7$ | *$69.9 \pm 7.0$* | $71.0 \pm 2.6$ |
| Ours-Aug | $25.9 \pm 40.5$ | $15.5 \pm 12.5$ | $20.9 \pm 13.8$ | $18.6 \pm 15.0$ | $4.7 \pm 2.1$ | $17.1 \pm 16.8$ |

Table 23: Out-of-domain accuracy on CAMELYON17 with nuclei on white background. The column name indicates the centre used to train models. The best accuracy for each column is in **bold face** and the second best in *italics*.

| Method | Centre-0 | Centre-1 | Centre-2 | Centre-3 | Centre-4 | Average |
|---|---|---|---|---|---|---|
| ERM | $73.1 \pm 7.9$ | $66.4 \pm 12.7$ | $50.0 \pm 0.0$ | $46.3 \pm 2.9$ | $49.5 \pm 4.0$ | $57.1 \pm 5.5$ |
| Macenko | $44.9 \pm 1.9$ | $63.5 \pm 10.4$ | $49.8 \pm 0.6$ | $45.4 \pm 2.0$ | $50.1 \pm 0.2$ | $50.7 \pm 3.0$ |
| RSC | $55.0 \pm 7.4$ | $61.4 \pm 10.0$ | $50.0 \pm 0.0$ | $48.0 \pm 2.9$ | $50.3 \pm 0.5$ | $52.9 \pm 4.2$ |
| L2D | $53.6 \pm 10.1$ | $84.1 \pm 1.8$ | $49.9 \pm 4.4$ | $41.6 \pm 8.9$ | $78.9 \pm 7.5$ | $61.6 \pm 6.5$ |
| Ours | *$90.1 \pm 1.0$* | **$92.9 \pm 0.4$** | *$89.9 \pm 1.9$* | **$92.9 \pm 0.3$** | $89.7 \pm 1.1$ | **$91.1 \pm 1.0$** |
| ERM-Aug | $58.6 \pm 8.1$ | $68.1 \pm 9.5$ | $56.9 \pm 6.9$ | $52.0 \pm 5.1$ | $54.8 \pm 6.6$ | $58.1 \pm 7.2$ |
| Macenko-Aug | $52.0 \pm 7.1$ | $82.7 \pm 9.1$ | $51.2 \pm 11.3$ | $51.2 \pm 5.7$ | $73.3 \pm 9.4$ | $62.1 \pm 8.5$ |
| RSC-Aug | $54.5 \pm 8.7$ | $75.9 \pm 2.8$ | $48.7 \pm 1.6$ | $51.0 \pm 5.0$ | $60.4 \pm 9.6$ | $58.1 \pm 5.5$ |
| L2D-Aug | $84.1 \pm 0.9$ | $89.0 \pm 0.6$ | $45.7 \pm 0.9$ | $67.5 \pm 7.6$ | **$90.3 \pm 1.5$** | $75.3 \pm 2.3$ |
| Ours-Aug | **$90.2 \pm 1.1$** | *$92.2 \pm 0.9$* | **$91.0 \pm 2.2$** | *$91.7 \pm 0.7$* | *$89.8 \pm 1.3$* | *$91.0 \pm 1.2$* |

Table 24: Out-of-domain accuracy on CAMELYON17 evaluated on binary nuclei segmentation masks. The column name indicates the centre used to train models. The best accuracy for each column is in **bold face** and the second best in *italics*.

| Method | Centre-0 | Centre-1 | Centre-2 | Centre-3 | Centre-4 | Average |
|---|---|---|---|---|---|---|
| ERM | $48.7 \pm 4.1$ | $50.0 \pm 0.0$ | $50.0 \pm 0.0$ | $50.0 \pm 0.0$ | $50.8 \pm 3.7$ | $49.9 \pm 1.6$ |
| Macenko | $50.0 \pm 0.0$ | $49.0 \pm 3.1$ | $49.6 \pm 1.1$ | $49.4 \pm 1.4$ | $50.0 \pm 0.0$ | $49.6 \pm 1.1$ |
| RSC | $50.0 \pm 0.0$ | $50.0 \pm 0.0$ | $49.6 \pm 1.3$ | $50.0 \pm 0.0$ | $50.0 \pm 0.0$ | $49.9 \pm 0.3$ |
| L2D | $59.4 \pm 10.5$ | $58.2 \pm 6.3$ | $52.7 \pm 5.0$ | $50.0 \pm 0.0$ | $50.2 \pm 0.7$ | $54.1 \pm 4.5$ |
| Ours | *$90.8 \pm 1.0$* | **$92.9 \pm 0.3$** | *$92.4 \pm 0.6$* | *$91.3 \pm 1.2$* | *$90.8 \pm 0.8$* | *$91.6 \pm 0.8$* |
| ERM-Aug | $49.3 \pm 1.9$ | $49.9 \pm 0.3$ | $50.0 \pm 0.0$ | $49.0 \pm 1.8$ | $49.7 \pm 0.6$ | $49.6 \pm 0.9$ |
| Macenko-Aug | $49.4 \pm 1.1$ | $50.3 \pm 1.0$ | $49.9 \pm 1.8$ | $48.2 \pm 1.1$ | $50.0 \pm 0.0$ | $49.6 \pm 1.0$ |
| RSC-Aug | $49.8 \pm 0.5$ | $50.0 \pm 0.0$ | $53.1 \pm 5.3$ | $50.0 \pm 0.0$ | $50.0 \pm 0.0$ | $50.6 \pm 1.2$ |
| L2D-Aug | $49.9 \pm 1.0$ | $58.4 \pm 5.8$ | $50.0 \pm 0.0$ | $50.8 \pm 2.4$ | $52.4 \pm 4.2$ | $52.3 \pm 2.7$ |
| Ours-Aug | **$91.6 \pm 0.5$** | *$92.4 \pm 0.9$* | **$92.5 \pm 1.0$** | **$92.4 \pm 0.4$** | **$91.4 \pm 0.4$** | **$92.1 \pm 0.7$** |

Table 25: Out-of-domain accuracy on CAMELYON17 evaluated on inverted nuclei segmentation masks. The column name indicates the centre used to train models. The best accuracy for each column is in **bold face** and the second best in *italics*.

| Method | Centre-0 | Centre-1 | Centre-2 | Centre-3 | Centre-4 | Average |
|---|---|---|---|---|---|---|
| ERM | $47.7 \pm 2.5$ | $50.8 \pm 2.5$ | $50.0 \pm 0.0$ | $47.3 \pm 1.9$ | $51.1 \pm 3.6$ | $49.4 \pm 2.1$ |
| Macenko | $45.3 \pm 1.6$ | $54.6 \pm 8.6$ | $49.9 \pm 0.4$ | $42.7 \pm 8.0$ | $50.0 \pm 0.0$ | $48.5 \pm 3.7$ |
| RSC | $48.7 \pm 1.9$ | $50.0 \pm 0.0$ | $50.0 \pm 0.0$ | $48.7 \pm 1.9$ | $50.0 \pm 0.0$ | $49.5 \pm 0.8$ |
| L2D | $43.5 \pm 4.8$ | $72.8 \pm 4.7$ | $57.1 \pm 9.2$ | $39.6 \pm 2.7$ | $48.6 \pm 5.1$ | $52.3 \pm 5.3$ |
| Ours | *89.0 ± 1.0* | **92.3 ± 0.4** | *90.6 ± 2.0* | *88.2 ± 2.6* | *86.5 ± 1.8* | *89.3 ± 1.6* |
| ERM-Aug | $48.2 \pm 2.0$ | $52.4 \pm 3.7$ | $53.2 \pm 8.9$ | $44.9 \pm 1.0$ | $49.4 \pm 0.7$ | $49.6 \pm 3.3$ |
| Macenko-Aug | $45.2 \pm 3.8$ | $76.0 \pm 13.4$ | $49.6 \pm 9.1$ | $46.1 \pm 0.9$ | $50.0 \pm 0.1$ | $53.4 \pm 5.5$ |
| RSC-Aug | $47.6 \pm 2.9$ | $51.8 \pm 10.6$ | $67.5 \pm 13.1$ | $44.9 \pm 1.7$ | $49.9 \pm 0.4$ | $52.3 \pm 5.7$ |
| L2D-Aug | $64.1 \pm 5.0$ | $87.5 \pm 1.2$ | $43.2 \pm 3.4$ | $58.7 \pm 11.0$ | $64.9 \pm 9.0$ | $63.7 \pm 5.9$ |
| Ours-Aug | **91.7 ± 0.5** | *91.7 ± 1.0* | **91.7 ± 1.4** | **92.7 ± 0.3** | **86.6 ± 2.3** | **90.9 ± 1.1** |

Table 26: Accuracy when combining L2D or RSC with the proposed method on CAMELYON17. The best accuracy for each column is in **bold face** and the second best in *italics*. "+Ours" results are an average of five models for each combination of medical centre and method instead of an average of ten models. All results are using photometric augmentations described in 4.3 even though "-Aug" is omitted from the name in the table.

| Method | Centre-0 | Centre-1 | Centre-2 | Centre-3 | Centre-4 | Average |
|---|---|---|---|---|---|---|
| L2D | **94.3 ± 0.1** | $87.6 \pm 0.6$ | $87.7 \pm 1.4$ | $83.4 \pm 2.6$ | **92.3 ± 0.9** | $89.1 \pm 1.1$ |
| L2D+Ours | $92.2 \pm 0.6$ | *92.8 ± 0.1* | *92.5 ± 0.5* | $84.8 \pm 1.5$ | $88.9 \pm 1.4$ | $90.3 \pm 0.8$ |
| RSC | *93.1 ± 0.8* | $78.2 \pm 2.0$ | $89.3 \pm 3.4$ | $77.9 \pm 2.2$ | $91.0 \pm 1.7$ | $85.9 \pm 2.0$ |
| RSC+Ours | $92.1 \pm 0.8$ | **93.6 ± 0.4** | *92.5 ± 1.2* | **91.3 ± 0.7** | *91.9 ± 0.4* | **92.3 ± 0.7** |
| Ours | $91.8 \pm 0.7$ | $92.2 \pm 1.6$ | **92.9 ± 0.7** | *90.4 ± 1.1* | $91.7 \pm 0.5$ | *91.8 ± 0.9* |

Table 27: Accuracy when combining L2D or RSC with the proposed method on BCSS. The best accuracy for each column is in **bold face** and the second best in *italics*. "+Ours" results are an average of five models for each combination of medical centre and method instead of an average of ten models. All results are using photometric augmentations described in 4.3 even though "-Aug" is omitted from the name in the table.

| Method | Centre-0 | Centre-1 | Centre-2 | Centre-3 | Centre-4 | Average |
|---|---|---|---|---|---|---|
| L2D | $81.9 \pm 0.3$ | $74.7 \pm 0.6$ | $74.2 \pm 0.6$ | $67.5 \pm 2.9$ | **77.2 ± 2.2** | $75.1 \pm 1.3$ |
| L2D+Ours | $81.8 \pm 2.0$ | *82.2 ± 0.5* | $73.7 \pm 1.3$ | $65.7 \pm 2.6$ | $73.8 \pm 2.2$ | $75.5 \pm 1.7$ |
| RSC | $79.6 \pm 1.5$ | $72.1 \pm 2.9$ | $71.6 \pm 1.7$ | $63.2 \pm 1.6$ | $74.1 \pm 4.3$ | $72.1 \pm 2.4$ |
| RSC+Ours | *82.0 ± 1.2* | **82.5 ± 1.7** | **76.7 ± 2.9** | *77.5 ± 2.0* | *75.8 ± 1.6* | **78.9 ± 1.9** |
| Ours | **82.3 ± 0.8** | $81.9 \pm 2.3$ | *75.7 ± 2.2* | **79.6 ± 1.0** | $74.8 \pm 1.9$ | *78.8 ± 1.6* |

Table 28: Accuracy when combining L2D or RSC with the proposed method on Ocelot. The best accuracy for each column is in **bold face** and the second best in *italics*. "+Ours" results are an average of five models for each combination of medical centre and method instead of an average of ten models. All results are using photometric augmentations described in 4.3 even though "-Aug" is omitted from the name in the table.

| Method | Centre-0 | Centre-1 | Centre-2 | Centre-3 | Centre-4 | Average |
|---|---|---|---|---|---|---|
| L2D | **74.7 ± 0.6** | $68.3 \pm 0.5$ | $65.6 \pm 0.7$ | *62.5 ± 2.7* | **74.4 ± 1.2** | *69.1 ± 1.1* |
| L2D+Ours | *72.2 ± 1.6* | $70.8 \pm 0.3$ | *69.4 ± 0.6* | $59.8 \pm 1.6$ | $69.9 \pm 2.3$ | $68.4 \pm 1.3$ |
| RSC | $71.9 \pm 2.2$ | $61.7 \pm 2.4$ | $69.2 \pm 2.9$ | $58.4 \pm 1.4$ | $70.1 \pm 3.6$ | $66.3 \pm 2.5$ |
| RSC+Ours | $71.1 \pm 0.7$ | **72.1 ± 0.6** | **69.6 ± 1.7** | **70.4 ± 1.7** | *70.9 ± 1.2* | **70.8 ± 1.2** |
| Ours | $70.8 \pm 0.5$ | *72.0 ± 1.3* | $68.9 \pm 1.3$ | **70.4 ± 0.7** | $70.7 \pm 1.3$ | *70.6 ± 1.0* |

Table 29: Accuracy when using a ViT-Tiny [60], comparing the baseline against the proposed method on CAMELYON17. The best accuracy for each column is in **bold face** and the second best in *italics*. All results are using photometric augmentations described in 4.3 even though "-Aug" is omitted from the name in the table. 'AwoC4' is 'Average without Centre-4'. 'RSNA' refers to RandStainNA [24].

| Method | Centre-0 | Centre-1 | Centre-2 | Centre-3 | Centre-4 | Average | AwoC4 |
|---|---|---|---|---|---|---|---|
| ERM | $94.6 \pm 0.5$ | $78.0 \pm 4.0$ | $90.0 \pm 1.6$ | $84.9 \pm 2.9$ | $87.7 \pm 2.5$ | $87.0 \pm 2.3$ | $86.9 \pm 2.3$ |
| RSNA | $94.6 \pm 0.4$ | *$86.8 \pm 2.0$* | $88.5 \pm 2.8$ | $84.9 \pm 2.3$ | *$90.6 \pm 2.9$* | $89.1 \pm 2.1$ | $88.7 \pm 1.9$ |
| DDCA | *$95.0 \pm 0.5$* | $77.9 \pm 3.7$ | $91.2 \pm 1.6$ | $81.7 \pm 4.1$ | $86.2 \pm 3.0$ | $86.4 \pm 2.6$ | $86.5 \pm 2.5$ |
| L2D | **$95.1 \pm 0.0$** | $81.0 \pm 0.1$ | *$91.8 \pm 0.8$* | *$87.3 \pm 0.8$* | **$91.8 \pm 0.1$** | *$89.4 \pm 0.4$* | *$88.8 \pm 0.4$* |
| Ours | $94.5 \pm 0.2$ | **$92.5 \pm 0.5$** | **$94.0 \pm 0.7$** | **$91.8 \pm 1.0$** | $86.7 \pm 2.1$ | **$91.9 \pm 0.9$** | **$93.2 \pm 0.6$** |

Table 30: Accuracy when using a ViT-Tiny [60], comparing the baseline against the proposed method on BCSS. The best accuracy for each column is in **bold face** and the second best in *italics*. All results are using photometric augmentations described in 4.3 even though "-Aug" is omitted from the name in the table. 'AwoC4' is 'Average without Centre-4'. 'RSNA' refers to RandStainNA [24].

| Method | Centre-0 | Centre-1 | Centre-2 | Centre-3 | Centre-4 | Average | AwoC4 |
|---|---|---|---|---|---|---|---|
| ERM | $79.2 \pm 3.3$ | $70.7 \pm 4.6$ | $68.6 \pm 2.4$ | $66.9 \pm 2.2$ | $69.5 \pm 5.1$ | $71.0 \pm 3.5$ | $71.4 \pm 3.1$ |
| RSNA | *$79.9 \pm 1.9$* | *$77.9 \pm 2.4$* | **$74.4 \pm 4.3$** | $72.1 \pm 2.4$ | *$75.2 \pm 5.4$* | *$75.9 \pm 3.3$* | *$76.1 \pm 2.8$* |
| DDCA | $78.1 \pm 2.7$ | $70.8 \pm 3.2$ | $69.9 \pm 4.5$ | $66.3 \pm 4.4$ | $65.8 \pm 4.6$ | $70.2 \pm 3.9$ | $71.3 \pm 3.7$ |
| L2D | **$80.8 \pm 0.1$** | $75.3 \pm 0.1$ | $71.6 \pm 4.1$ | **$75.9 \pm 0.3$** | **$79.1 \pm 0.3$** | **$76.5 \pm 1.0$** | $75.9 \pm 1.2$ |
| Ours | $78.6 \pm 1.1$ | **$79.5 \pm 1.2$** | *$74.3 \pm 2.4$* | *$75.4 \pm 1.3$* | $61.9 \pm 3.6$ | $74.0 \pm 1.9$ | **$77.0 \pm 1.5$** |

Table 31: Accuracy when using a ViT-Tiny [60], comparing the baseline against the proposed method on Ocelot. The best accuracy for each column is in **bold face** and the second best in *italics*. All results are using photometric augmentations described in 4.3 even though "-Aug" is omitted from the name in the table. 'AwoC4' is 'Average without Centre-4'. 'RSNA' refers to RandStainNA [24].

| Method | Centre-0 | Centre-1 | Centre-2 | Centre-3 | Centre-4 | Average | AwoC4 |
|---|---|---|---|---|---|---|---|
| ERM | *$71.7 \pm 1.9$* | $61.0 \pm 3.5$ | $67.1 \pm 3.1$ | $62.6 \pm 3.5$ | $67.3 \pm 4.3$ | $65.9 \pm 3.2$ | $65.6 \pm 3.0$ |
| RSNA | $71.2 \pm 1.8$ | *$68.6 \pm 2.8$* | $67.2 \pm 2.5$ | $68.3 \pm 1.7$ | *$69.2 \pm 3.6$* | *$68.9 \pm 2.5$* | $68.8 \pm 2.2$ |
| DDCA | **$73.0 \pm 1.8$** | $61.2 \pm 2.0$ | $67.2 \pm 0.9$ | $61.6 \pm 4.2$ | $62.6 \pm 4.1$ | $65.1 \pm 2.6$ | $65.7 \pm 2.2$ |
| L2D | $70.8 \pm 0.1$ | $65.9 \pm 0.1$ | *$68.9 \pm 1.0$* | **$71.8 \pm 0.7$** | **$71.5 \pm 0.3$** | **$69.8 \pm 0.4$** | *$69.3 \pm 0.5$* |
| Ours | $69.3 \pm 0.8$ | **$68.8 \pm 0.8$** | **$70.0 \pm 0.6$** | *$70.4 \pm 1.0$* | $64.5 \pm 2.3$ | $68.6 \pm 1.1$ | **$69.6 \pm 0.8$** |

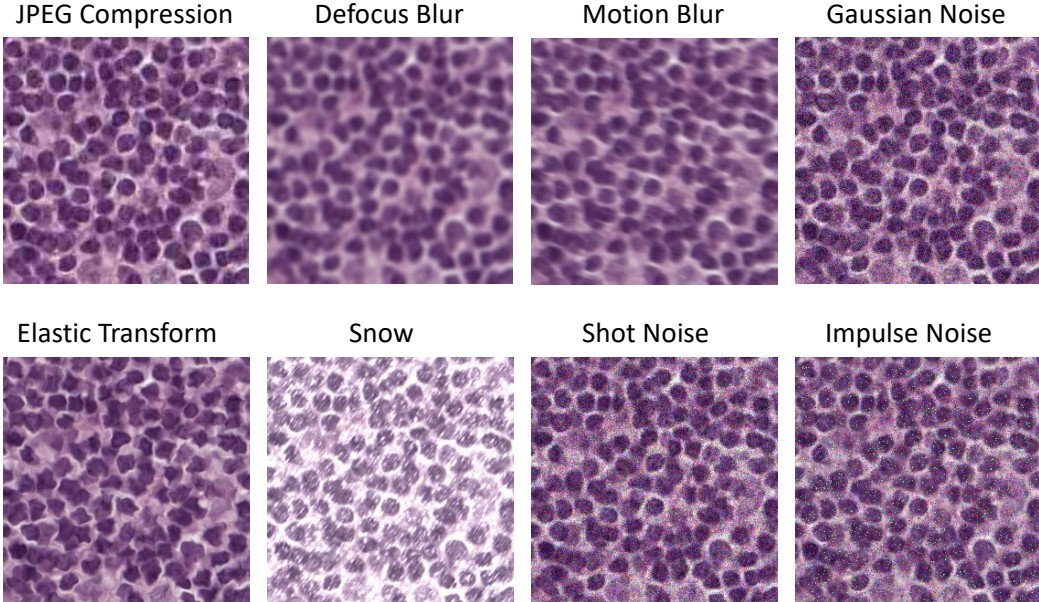

Figure 5: Exemplary image corruptions from [61] applied to an input image used in this study.

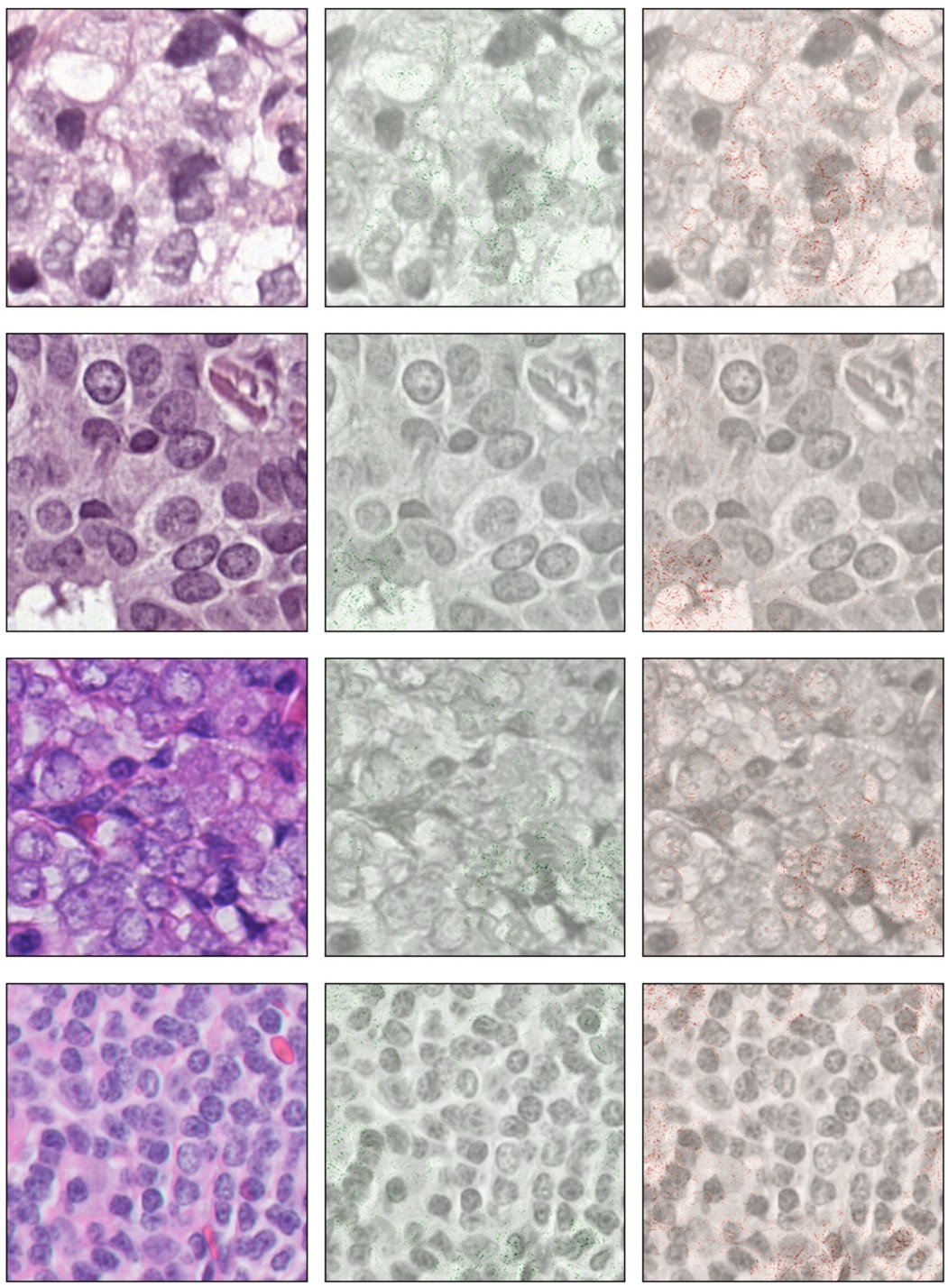

Figure 6: Saliency maps for four randomly selected tiles using Integrated Gradients [64] for a model trained via ERM-Aug. The green-coloured map indicates the contribution towards the positive class (tumour), and the red one towards the negative class (non-tumour).

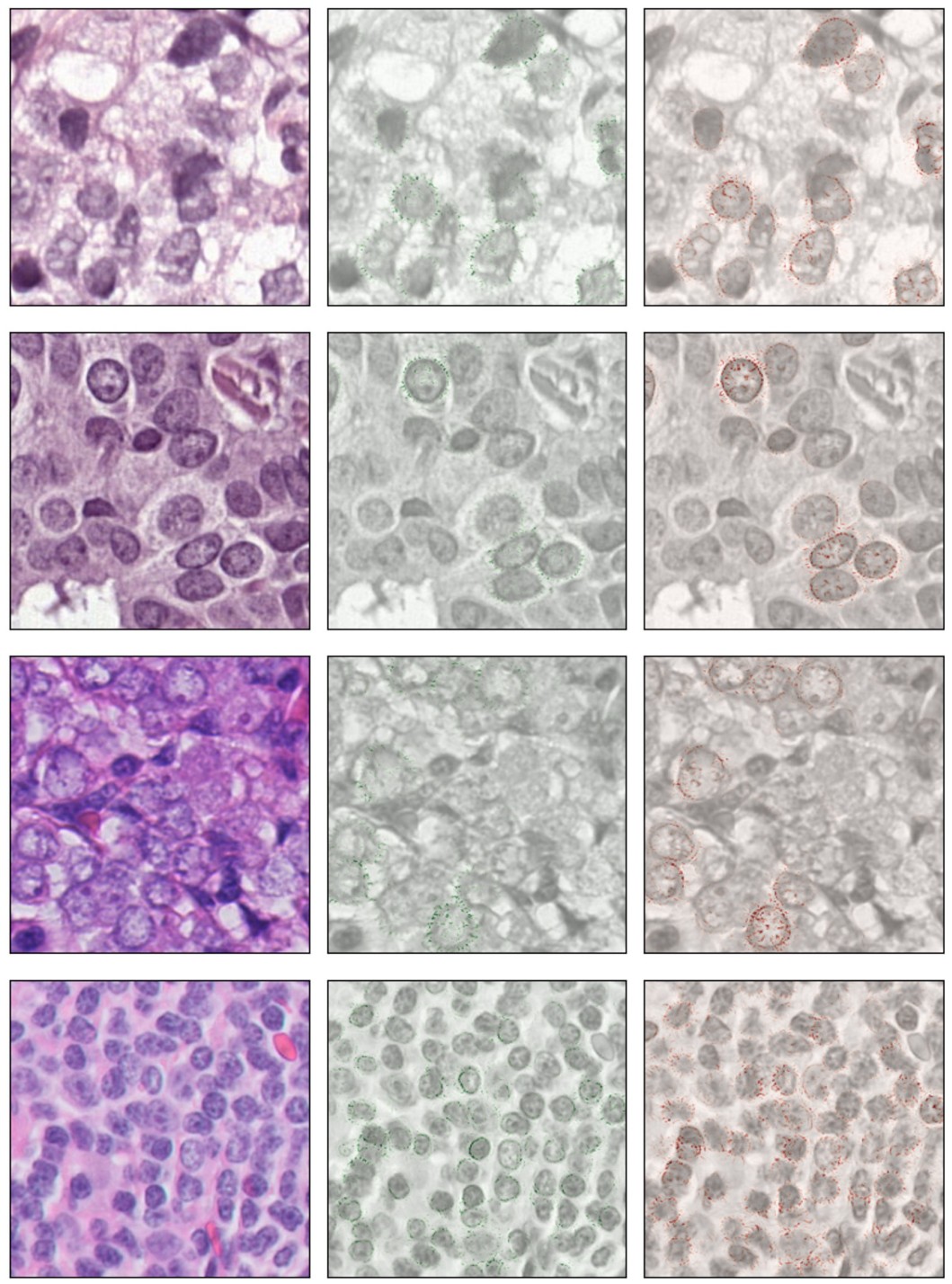

Figure 7: Saliency maps for four randomly selected tiles using Integrated Gradients [64] for a model trained via Ours-Aug. The green-coloured map indicates the contribution towards the positive class (tumour), and the red one towards the negative class (non-tumour).

