# OpenReview forum: "Are nuclear masks all you need for improved out-of-domain generalisation? A closer look at cancer classification in histopathology"
_NeurIPS.cc/2024/Conference — NeurIPS 2024 poster_

### Official Review · Reviewer_sg4y · 2024-07-05

**Soundness:** 2
**Presentation:** 2
**Contribution:** 2
**Rating:** 5
**Confidence:** 5

**Summary:**

This paper tackles the issue of out-of-domain (OOD) generalization in histopathology, which is complicated by domain shifts due to different scanners, staining procedures, and inter-patient variability. With a focus on single-domain generalization, this paper proposes to prioritize shape features over texture, as shape remains consistent across domains. This is achieved by using images and segmentation masks with data augmentation and regularization, emphasizing nuclear morphology critical for cancer detection. Experiments confirm this approach improves OOD generalization, validating an old hypothesis with deep learning and demonstrating the method's flexibility as an enhancement to other techniques.

**Strengths:**

- Vulnerability to cross-domain variances and shifts is essentially a critical challenge in the context of AI-driven digital pathology. Diving deeper into this topic could greatly enhance the reliability and usability of current learning-based approaches.

- The experimental evaluations and ablation studies are quite comprehensive.

**Weaknesses:**

- The paper is overall poorly structured with convoluted paragraph organizations and intricate sentences. It is hard for readers to get the core idea of this paper.

- The description of proposed methodology is quite informal, with ambiguous elucidation and unrigorous mathematical formulations. For example, what do you mean by stating “run a forward pass”?

- The idea to leverage the shape of nuclei as domain-invariant markers is not new [Ref-A]. The morphology and spatial distribution of nucleus clusters are also common biomarkers for cancer diagnosis and grading [Ref-B]. It is unclear what the contribution of this work is.

- Why not directly employing a segmentation to extract the morphometrics for all nuclei and then input the secured measurements to a simple classifier (e.g., SVM) to get the diagnosis prediction? If the color and texture are argued to be not robust against domain shifts, why not simply discard the original pathology slides?

[Ref-A]:
Sharma, Yash, Sana Syed, and Donald E. Brown. "Mani: Maximizing mutual information for nuclei cross-domain unsupervised segmentation." in MICCAI, 2022.

[Ref-B]:
Bansal, Cherry, et al. "Grading systems in the cytological diagnosis of breast cancer: a review." Journal of cancer research and therapeutics 10.4 (2014): 839-845.

**Questions:**

- Could you carefully revise the manuscript to make its content self-organized and easy to follow?

- Compared to the suggested alternative solutions, what is the advantage of adopting the proposed approach? In-depth investigation and analysis are required in this respect.

**Limitations:**

The authors have clearly identified the limitations of their work and suggested sound solutions. No potential negative societal impact.

---

> ### Author Rebuttal · Authors · 2024-08-06
>
> Thank you for assessing our paper and recognizing the importance of domain shifts in AI-driven digital pathology and the comprehensiveness of our experimental evaluations and ablation studies.
>
> We are committed to revise the paper so that it is better organized and easier to follow with a more accessible language, clarification or omission of elucidations that might be ambiguous, and the use of more rigorous mathematical formulations. Regarding the latter, we consider it beneficial to define more formally which embeddings $z$ and $z'$ are (it is the embedding just before the Global Average Pooling in ResNet-50 of the input image $x$ and its corresponding segmentation mask $x'$, respectively) and what predictions $\hat{y}$ and $\hat{y}'$ are (it is the final output of the model for $x$ and $x'$, respectively). By stating "run a forward pass", we meant to execute a forward propagation through the neural network model, referencing the forward method used in standard code and textbooks (e.g., http://d2l.ai/chapter_linear-regression/oo-design.html#models). We will revise the formulations in the paper accordingly. If there are any other mathematical formulations that might be perceived as unrigorous, we kindly ask the reviewer to specify them concretely via an official comment.
>
> As for the more technical comments, we acknowledge that utilizing the shape and organization of nuclei is not new in machine learning in general and deep learning in particular. However, we did not make such a claim in the paper but rather referenced multiple earlier contributions, please see line 57 through 63 in the paper. As noted there, an essential difference between the approach we propose and previous approaches is that our approach enables the training of neural network models that robustly analyze the original images directly, while many other approaches uses the segmentation masks also at inference.
>
> Also the paper mentioned in [Ref-A] does not propose an approach similar to ours. It rather deals with segmentation of nuclei and propose an architecture including a backbone encoder-decoder network, a segmentation head, a projection head, and a network combining the outputs from the heads and estimating the mutual information. Our paper deals with tumor classification and simply adds a loss and augments the input during training. As such, these approaches are not comparable in terms of task and complexity. It is interesting to note that [Ref-A] uses HoVer-Net as the backbone encoder-decoder network. Based also on the comments from reviewer UomK, we have now performed experiments with fine-tuning HoVer-Net for tumor classification. As seen in the PDF attached to the "global" response (see Tables 1 to 3 in that PDF), our approach compares favorably also in this case.
>
> Regarding [Ref-B], we fully agree that the importance of nuclear morphology is well recognized in the medical literature. Indeed, the knowledge about this and also the importance of organization of nuclei in the tissue is part of the motivation for using nuclear segmentation masks in our proposed approach. Line 45 through 51 in our paper addresses this with multiple references, including reference 9 that defines perhaps the most famous grading system in breast cancer histopathology (which is also cited within [Ref-B]). If you recommend to cite also [Ref-B] to cover insights from breast cancer cytology in addition, we are happy to do so (although please note that our paper is about histopathology).
>
> We hope that these explanations have made the contribution of our work clear and will in the revision of the paper make sure that this is expressed clearly. If you think otherwise or have additional concerns, we kindly ask you to specify them concretely.
>
> Because our study aims to improve an aspect of neural networks, we do not consider it to be relevant to compare the performance of our approach with that of classical feature extraction followed by e.g. SVM. We consider it to be shown that neural networks can perform better than classical approaches for the task studied in our paper (tumor classification) and many other tasks in histopathology and beyond. As a concrete example for tumor classification, the results of the CAMELYON16 challenge show that the submitted algorithm using several nucleus-based features and 6 other submitted algorithms extracting features and applying a supervised classification method perform substantially worse than the 25 submitted algorithms using deep learning (see Table 2 in https://jamanetwork.com/journals/jama/fullarticle/2665774). We have therefore chosen to not evaluate the performance of using morphometrics for all nuclei and a simple classifier (e.g., SVM), as this is neither expected to give as good results nor would it provide neural network models that robustly analyze the original images directly.

---

> > ### Comment · Reviewer_sg4y · 2024-08-12
> > **Thank the authors for their detailed responses**
> >
> > Thank the authors for their detailed responses. Their efforts for revising the presentation of the manuscript are appreciated. Their clarifications on technical novelty also resolve some of my concerns. I would therefore increase my score to 5, marginally above the acceptance threshold.

---

### Official Review · Reviewer_fXPp · 2024-07-08

**Soundness:** 2
**Presentation:** 2
**Contribution:** 3
**Rating:** 5
**Confidence:** 4

**Summary:**

The paper proposes a method focusing on nuclei masks, along with an augmentation method, to train a feature extractor to increase
the out-of-domain generalization by directing the model toward learning nuclear features.
The method has been evaluated against 4 different approaches on three different cancer datasets comprised of different centers. It has been empirically shown that the nuclei-based approach is superior to other methods, and also the augmentation method can improve the performance of different approaches when used as a plugin.

**Strengths:**

The methodology proposed is simple and straightforward and appears to be superior to other methods using a resnet50 backbone. There are different ablation studies supporting the authors' claims for the components of their design. Also, generally, the idea is biologically interesting to lead the model to focus on nuclei.

**Weaknesses:**

major:
The methods used to benchmark against are not the recent methods (with the newest method (L2D) being from 2021). This needs to be addressed by comparing it against some new pathology-specific methods like (1), (2), or (3).

The authors only used resnet50 for most of their experiments and used vit_tiny for limited experiments against ERM. In order to show the generality of the method, vit_tiny needs to be added to the full set of experiments, along with other methods for benchmarking.

(1) Marini, Niccolò, et al. "Data-driven color augmentation for H&E stained images in computational pathology." Journal of Pathology Informatics 14 (2023): 100183.

(2) Shen, Y., Luo, Y., Shen, D., Ke, J. (2022). RandStainNA: Learning Stain-Agnostic Features from Histology Slides by Bridging Stain Augmentation and Normalization. In: Wang, L., Dou, Q., Fletcher, P.T., Speidel, S., Li, S. (eds) Medical Image Computing and Computer Assisted Intervention – MICCAI 2022. MICCAI 2022. Lecture Notes in Computer Science, vol 13432. Springer, Cham. https://doi.org/10.1007/978-3-031-16434-7_21

(3) Bouteldja, Nassim, et al. "Stain-independent deep learning–based analysis of digital kidney histopathology." The American Journal of Pathology 193.1 (2023): 73-83.

minor:
the contributions of the work in the introduction can be written more clearly: it is a bit vague to understand the key points through the first read :)

**Questions:**

The work has been built upon the idea that the network should focus on nuclei to generalize better. Yet, I would suggest the authors add a qualitative experiment visualizing the saliency map of resnt50 (or attention map of vit) to support indeed that the focus on nuclei is the key to the improvement. This can be compared with some sort of human annotation and also against other methods.

---

> ### Author Rebuttal · Authors · 2024-08-06
>
> Thank you for recognizing the simplicity yet superiority of our approach as well as its biological appeal. We also appreciate your suggestions for additional analyses, which add to our previous experiments to show the strengths of our proposed approach.
>
> > The methods used to benchmark against are not the recent methods (with the newest method (L2D) being from 2021). This needs to be addressed by comparing it against some new pathology-specific methods like (1), (2), or (3).
>
> We were able to complete analyses for the approaches described in your reference (1) and (2). Please find the results in Tables 1, 2, and 3 in the PDF attached to the "global" response. These results demonstrate that our approach compares favorably also to these recent pathology-specific methods, thus adding support for the superiority of our approach.
>
> > The authors only used resnet50 for most of their experiments and used vit_tiny for limited experiments against ERM. In order to show the generality of the method, vit_tiny needs to be added to the full set of experiments, along with other methods for benchmarking.
>
> We have substantially expanded on these experiments. Although results are not yet generated for the baseline methods that performed poorly using ResNet-50 (specifically no augmentation, Macenko normalization, and RSC), we managed to expand the experiments to include the other methods and even include the new pathology-specific methods you suggested in reference (1) and (2) (although for DDCA, the method in your reference (1), the number of models trained per center was between 5 and 10 instead of always 10 because of time constraints; we will amend this during the discussion phase but are very confident that this will not change the interpretation of the results). Please find the results in Tables 4 to 6 in the PDF attached to the "global" response. These results show that our approach obtains superior out-of-domain performance when classifying the same cancer type in different centers and scanners for the CAMELYON17 dataset. When classifying tumors in other cancer types, the results are more mixed. In particular, it seems that models trained on one of the five centers (Center-4) in CAMELYON17 do not generalize well to other cancer types and are actually also performing sub-optimally in the four other centers in CAMELYON17. For models trained on each of the four other centers in CAMELYON17, the performance is, on average, better than other approaches also in new cancer types, but the performance increase is lower than for ResNet-50. However, in the same cancer type (CAMELYON17 data), the performance gain is similar for both ViT tiny and ResNet-50. Our interpretation of all these results is that our approach can improve out-of-domain performance also for ViT tiny, in particular across centers and scanners for the same cancer type, but that it might also fail for a minority of the training datasets. We have not had time to investigate the reason for this occasional failure further.
>
> > minor: the contributions of the work in the introduction can be written more clearly: it is a bit vague to understand the key points through the first read :)
>
> Thank you for the feedback. We are committed to revise the introduction and use a more accessible language that will make it easier to understand the key points and contributions of our work.
>
> > The work has been built upon the idea that the network should focus on nuclei to generalize better. Yet, I would suggest the authors add a qualitative experiment visualizing the saliency map of resnt50 (or attention map of vit) to support indeed that the focus on nuclei is the key to the improvement. This can be compared with some sort of human annotation and also against other methods.
>
> Thank you for the suggestion. We have now generated some saliency maps of ResNet-50 models using the Integrated gradients method, and they clearly highlight nuclei for our method while not for others. The generated attribution maps highlight the outline of some nuclei in the images for our method, while the attribution is dispersed in a region for ERM. We did not include examples in the rebuttal PDF because of insufficient space but will include the images in our paper. We will also compare these images to segmentations of nuclei in order to confirm qualitatively that the models trained using our approach focus more on nuclei. Please also note that multiple of the ablation studies reported in our paper indicate that models trained using our method focus more on nuclei, in particular those commented in line 244 through 283 with results in Tables 13 to 26, but also the results in Tables 27 and 28 support this.

---

> > ### Comment · Reviewer_fXPp · 2024-08-08
> > **A question regarding the new experiments**
> >
> > Thank you for adding the new experiments and providing further details. I have a couple of questions and would appreciate further details on them from the authors.
> >
> > About this observation: "In particular, it seems that models trained on one of the five centers (Center-4) in CAMELYON17 do not generalize well to other cancer types and are actually also performing sub-optimally in the four other centers in CAMELYON17"
> > I am wondering how the distribution of samples in C4 differs from the samples in other centers. This can be done by reporting the number of samples per class per center.
> >
> > Also, I might have missed it but I could not find detailed explanations in the paper to mention if the reported ACC is patch-level or slide-level. If it is slide-level, which aggregation technique has been used? and if it is patch-level, I would like to hear the rationale behind it.

---

> > > ### Author Response · Authors · 2024-08-10
> > > **Dataset statistics and possible hypothesis for poor performance when training on C4**
> > >
> > > > I am wondering how the distribution of samples in C4 differs from the samples in other centers. This can be done by reporting the number of samples per class per center.
> > >
> > > Thank you for asking. There are aspects of the datasets that did not appear relevant for the ResNet-50 experiments but might possibly explain the difference for Center-4 with ViT. The number and distribution of patches across centres and patients are:
> > >
> > > Centre 0:
> > >
> > > Training subset with 44693 non-tumour patches and 44621 tumour ones from 5 patients. The percentage of no-tumor&tumor patches each patient contributed was 19.54%&0.13%, 19.27%&1.2%, 4.3%&56.9%, 15.03%&0.02%, 41.85%&41.75%
> > >
> > > Validation subset with 14100 non-tumor patches and 14172 tumour ones from 2 patients. The percentage of no-tumor&tumor patches each patient contributed was 38.33%&15.02%, 61.67%&84.98%
> > >
> > > Centre 1:
> > >
> > > Training subset with 24474 non-tumour patches and 25755 tumour ones from 6 patients. The percentage of no-tumor&tumor patches each patient contributed was 32.2%&25.9%, 11.84%&1.09%, 12.82%&65.99%, 16.07%&4.61%, 7.98%&1.24%, 19.09%&1.17%
> > >
> > > Validation subset with 9048 non-tumour patches and 7767 tumour ones from 2 patients. The percentage of no-tumor&tumor patches each patient contributed was 82.04%&0.91%, 17.96%&99.09%
> > >
> > > Centre 2:
> > >
> > > Training subset with 66835 non-tumour patches and 65519 tumour ones from 7 patients. The percentage of no-tumor&tumor patches each patient contributed was 10.97%&0.07%, 11.8%&0.02%, 8.28%&12.42%, 21.75%&0.53%, 19.23%&4.3%, 13.64%&0.32%, 14.33%&82.34%
> > >
> > > Validation subset with 17373 non-tumour patches and 18689 tumour ones from 2 patients. The percentage of no-tumor&tumor patches each patient contributed was 27.93%&29.21%, 72.07%&70.79%
> > >
> > > Centre 3:
> > >
> > > Training subset with 99268 non-tumour patches and 97137 tumour ones from 7 patients. The percentage of no-tumor&tumor patches each patient contributed was 21.51%&0.03%, 7.78%&0.03%, 7.25%&0.32%, 12.36%&0.3%, 21.76%&0.02%, 10.52%&0.09%, 18.81%&99.21%
> > >
> > > Validation subset with 30195 non-tumor patches and 32326 tumour ones from 3 patients. The percentage of no-tumor&tumor patches each patient contributed was 20.71%&0.75%, 42.82%&13.24%, 36.47%&86.01%
> > >
> > > Centre 4:
> > >
> > > Training subset with 111010 non-tumour patches and 108346 tumour ones from 7 patients. The percentage of no-tumor&tumor patches each patient contributed was 12.52%&0.2%, 19.69%&0.17%, 27.66%&0.02%, 13.51%&0.04%, 11.25%&0.65%, 8.41%&0.08%, 6.96%&98.84%
> > >
> > > Validation subset with 29211 non-tumour patches and 31875 tumour ones from 2 patients. The percentage of no-tumor&tumor patches each patient contributed was 56.78%&37.31%, 43.22%&62.69%
> > >
> > > Thus, Centre-4 is the only centre where nearly all (close to 99%) of the tumour patches used for training come from a single patient at the same time as less than 10% of the non-tumour patches come from that patient. Such imbalance in the training data might encourage learning of patient-specific features in some cases. A hypothesis is that such overfitting caused the suboptimal performance when training on Centre-4 with our approach and ViT.
> > >
> > > > Also, I might have missed it but I could not find detailed explanations in the paper to mention if the reported ACC is patch-level or slide-level. ... I would like to hear the rationale behind it.
> > >
> > > The WILDS package provides access to a subset of CAMELYON17 that consists of 10 slide images per centre, each of which "was manually annotated with tumour regions by pathologists" (reference 60 in our paper). Our analyses of CAMELYON17 are based on these 50 slide images. Since this implies that all analyzed slide images contain tumour, slide-level accuracy is not a good metric in this case (there is only one true class). We therefore used patch-level accuracy, which is also used in other papers utilising the WILDS package (see references 1,2,3 below) and in some papers not utilising the WILDS package (e.g. reference 4 below).
> > >
> > > Please note that we split into training and validation subsets such that all patches from a patient were in either training or validation, never in both (as commented in lines 169-171). This prevents the performance on the validation subset from being affected by patient-specific features.
> > >
> > > (1) Robey, Alexander, George J. Pappas, and Hamed Hassani. "Model-based domain generalization." Advances in Neural Information Processing Systems 34 (2021): 20210-20229.
> > >
> > > (2) Zhang, Zheyuan, et al. "Domain generalization with correlated style uncertainty." Proceedings of the IEEE/CVF Winter Conference on Applications of Computer Vision. 2024.
> > >
> > > (3) Chen, Junming, et al. "Federated domain generalization for image recognition via cross-client style transfer." Proceedings of the IEEE/CVF Winter Conference on Applications of Computer Vision. 2023.
> > >
> > > (4) Stacke, Karin, et al. "Measuring domain shift for deep learning in histopathology." IEEE journal of biomedical and health informatics 25.2 (2020): 325-336.

---

> > > > ### Comment · Reviewer_fXPp · 2024-08-12
> > > >
> > > > I thank the authors for providing further details. The authors' response has addressed my concerns. As I pointed out before, the writing can be improved (the reviewer UomK mentioned that too). Considering this, and also the new set of results provided, I would increase my rating to a Borderline Accept.

---

> > > > > ### Author Response · Authors · 2024-08-13
> > > > > **We thank the reviewers**
> > > > >
> > > > > We thank the reviewers for their feedbacks and revising their scores. If given the chance, we will thoroughly revise the writing and presentation in our paper before submitting the camera ready version, which would also include the new results that addressed the reviewers' concerns.

---

### Official Review · Reviewer_UomK · 2024-07-20

**Soundness:** 1
**Presentation:** 2
**Contribution:** 2
**Rating:** 5
**Confidence:** 3

**Summary:**

The paper "Are nuclear masks all you need for improved out-of-domain generalization? A closer look at cancer classification in histopathology." approaches the task of center classification for out-of-domain histopathology imagery. Motivated by classical shape-based segmentation, the authors suggest utilizing segmentation maps as an input to force the classification network to focus more on the morphology of cells. This enables the classification network to achieve a more robust classification accuracy on out-of-domain histopathology images.

**Strengths:**

The presented approach is certainly interesting, using segmentation maps to shift the attribution/focus of the classification network to cells. While I'm not too familiar with the recent SOTA approach in this domain, this approach seems to be novel. Using the segmentation mask leads to an increased classification accuracy on OOD data.

**Weaknesses:**

The paper lacks clarity; I often find myself rereading sentences and paragraphs to follow the ideas of the authors. For instance, the abstract does not clearly state which specific tasks the authors are approaching.

Many statements in the introduction motivating the presented approach are not supported by citations. An example include the last sentence of the second paragraph.

Additionally, the related work section mainly provides an enumeration of existing approaches and does not clearly explain how the presented approach fits/extends related work.

My main concern is the use of more extensive annotations, namely segmentation maps. The presented approach uses the predictions of HoVer-Net that rely on dense segmentation labels. Subsequentially, the authors require substantially more supervision than the approaches they compare against (e.g., L2D). Providing a discussion on this is required. Additionally, a more competitive baseline also using mask supervision would be required. For instance, training a network on segmentation and then fine-tuning it on the downstream task.

Further, the presented method does not consistently outperform L2D when data augmentation is utilized while using extensively more supervision.

As the main aim of using segmentation maps is to shift the attribution of the classification network, the authors should also discuss the connection to attribution methods or consider analyzing the attribution of their approach and other approaches.

Finally, many tables presented both in the supplement and the main paper do not adhere to the style guidelines and use too much horizontal space.

Minor comments:
- Line 17: Labeling noise is not really changing the data distribution.
- Line 29: What does general vision refer to? I think the domain of natural images is way more concise here.
- Line 51: “directing the attention of CNNs” sounds somewhat misleading. I guess the attribution is relevant in this case.
- Line 130: PyTorch does not provide a ResNet-50 implementation. The authors probably refer to TorchVision.
- Line: 31 is -> are.

**Questions:**

Does the slight increase in classification performance justify the use of significantly more extensive supervision?
How does the attribution change when utilizing the presented approach?
How complimentary is the presented approach to other existing approaches? Can I combine L2D and your approach?

**Limitations:**

The authors have not adequately addressed the limitations of their approach. As mentioned in my main concern, the presented method requires significantly more supervision than other approaches. This is not discussed in the limitations sections.

---

> ### Author Rebuttal · Authors · 2024-08-06
>
> Thank you for your comments and for recognizing our approach as interesting and novel.
>
> We are committed to revise the paper to use more accessible language and make it easier to understand our ideas and statements. This will include specifying clearly in the abstract that we analyze cancer classification across different centers, whole-slide scanners, and cancer types. We will also make it clearer in the related work section that our approach is different from other S-DG methods in that it attempts to bias the model to focus on nuclei specifically, or more generally towards some domain-general features captured by an image map, without the need to have the segmentation or image map during inference. We will also mention the vague resemblance to contrastive learning, although our approach is with supervision and jointly uses a classification loss, which also implies that we do not need nor use negative examples as in contrastive learning. Additionally, we will include citations like (1) and (2) that survey domain generalization approaches for histopathology. Both surveys indicate that S-DG methods developed for natural images have not been widely adopted for histopathology. Histopathology has instead seen its own S-DG methods developed over time, most of which are about augmentation or normalization. We will amend the introduction accordingly and also address the minor concerns you raised, including adjusting the style of the tables to adhere to the guidelines.
>
> (1) Jahanifar, Mostafa, et al. "Domain generalization in computational pathology: survey and guidelines." arXiv preprint arXiv:2310.19656 (2023).
>
> (2) Yoon, Jee Seok, et al. "Domain generalization for medical image analysis: A survey." arXiv preprint arXiv:2310.08598 (2023).
>
> > "My main concern is the use of more extensive annotations, namely segmentation maps. (...) Providing a discussion on this is required. Additionally, a more competitive baseline also using mask supervision would be required. For instance, training a network on segmentation and then fine-tuning it on the downstream task."
>
> We agree that our approach requires more information during training because it utilizes segmentation masks, although these are not needed at inference time. To investigate whether it is this additional information during training that improves performance, we have, as you suggested, performed additional experiments with a baseline that also builds on the segmentations. Specifically, we fine-tuned the ResNet-50 model in HoVer-Net (the segmentation model used to create the segmentation masks) on our downstream task using the same setup as our approach. As seen in Tables 1 to 3 in the PDF attached to the "global" response, our approach compares favourably also in this case, demonstrating that our approach utilizes the segmentations and the original images in a more efficient manner in order to attain high out-of-domain accuracy. Please also note that the experiments commented on lines 234 through 243, with associated results in the two bottom rows of Table 1 in the paper, all use the segmentation masks as in our approach but without the $l_2$-regularization. These results also suggest that simply including the segmentations in training is not important to the performance; it needs to be coupled with the $l_2$-regularization that pulls the feature representation of original images towards the feature representation of the segmentation mask. However, we will expand the limitation paragraph to mention that if our approach is to be applied to other fields than histopathology, it might be more difficult to generate image maps that capture domain-invariant features, in which case our approach might not be applicable due to the requirement of such segmentation maps.
>
> > "Further, the presented method does not consistently outperform L2D when data augmentation is utilized while using extensively more supervision."
>
> Although L2D occasionally performs slightly better than our approach when training on specific centres, our approach is clearly more accurate on average. The models trained using our approach are also more robust than models trained using L2D; please see additional results supporting this finding in the "global" response (in particular, Figure 1 in the PDF attached to that response).
>
> > "As the main aim of using segmentation maps is to shift the attribution of the classification network, the authors should also discuss the connection to attribution methods or consider analyzing the attribution of their approach and other approaches."
>
> Thank you for the suggestion. We have now generated some saliency maps using the Integrated gradients method, and they clearly highlight nuclei for our method while not for others. The generated attribution maps highlight the outline of some nuclei in the images for our method, while the attribution is dispersed in a region for ERM. We did not include examples in the rebuttal PDF because of insufficient space, but will include the images in our paper. Please also note that multiple of the ablation studies reported in our paper indicate that models trained using our method focus more on nuclei, in particular, those commented in lines 244 through 283 with results in Tables 13 to 26, but also the results in Tables 27 and 28 support this.
>
> > "How complimentary is the presented approach to other existing approaches? Can I combine L2D and your approach?"
>
> Please see lines 284 through 289 and Tables 29 to 31. Combining with L2D does not yield better performance than our approach alone, likely due to the opposing effects of these two interventions. Combining with RSC yielded a small gain over our approach alone. Overall, this demonstrates the effectiveness of the method proposed in this paper. It might also suggest that even better performance could be obtained by combining our method with a method that improves out-of-domain generalization more substantially than RSC while not having opposing effects as our method.

---

> > ### Comment · Reviewer_UomK · 2024-08-12
> > **Response Rebuttal**
> >
> > I thank the authors for their rebuttal and additional requirements. I'm happy to raise my score to 5 based on the additional additional comparisons. However, I still have doubts about the overall quality (presentation, soundness) of the paper; thus, I'm not able to raise my score higher.

---

### Author Rebuttal · Authors · 2024-08-06

We would like to thank the reviewers for recognizing the approach we propose as interesting, important, and novel. We also appreciate their input on the language, organization, and clarity of statements in our paper, which we will use to revise the paper thoroughly to make it easier to follow. The reviewers also provided useful comments that guided us to perform additional ablation studies that also demonstrate the superiority of our approach, for which we are most grateful. The most important results from the additional analyses are provided in the PDF attached to this comment. Please find our answers to each reviewer below with references to this PDF. After revising the paper to include these additional analyses and to improve the readability, we hope the paper is considered suitable for NeurIPS.

Tables 1 to 6 in the attached PDF present the results of the new experiments requested by the reviewers. Additionally, Figure 1 in the attached PDF provides further support for the increased robustness of models trained using our approach, thus complementing the results in Tables 4 to 6 in our paper. This figure shows the results of testing the robustness of ERM, L2D, and our method against adversarial attacks using the PGD attack. More specifically, Figure 1(a) shows the robustness of the different methods against the attack in the usual way, while Figure 1(b) shows the results of cross-model-attacks. For cross-model-attacks, we generate adversarial images using models from method A and then test the accuracy of models from methods B and C on those images, i.e., a method generates adversarial images for other methods. These results clearly show that adversarial images generated via models trained using our method are able to successfully attack the other models, and that adversarial images generated via models trained using the other methods have only a very small impact on the accuracy of models trained using our method.

---

### Decision · Program_Chairs · 2024-09-25

**Decision:**

Accept (poster)

**Comment:**

The paper addresses the challenge of out-of-domain generalization in computational histopathology by focusing on nuclei in images, proposing a novel training method that mixes original images with segmentation masks. The approach enhances the model's robustness to variations in staining procedures and imaging equipment, common across different hospitals. Reviewers appreciated the method's simplicity and broad applicability. However, there were concerns regarding the paper's overall presentation and clarity, particularly about the detailed explanation of results and the rationale behind certain methodological choices. The authors' rebuttal, which included additional comparisons and clarifications, addressed many of these concerns, leading reviewers to raise their scores marginally. While the paper still has some issues related to writing and clarity, the consensus leans toward acceptance.